# Plasma sphingolipid abnormalities in neurodegenerative diseases

Hideki Oizumi[1], Yoko Sugimura[1], Tomoko Totsune[1], Iori Kawasaki[1], Saki Ohshiro[1], Toru Baba[1], Teiko Kimpara[1], Hiroaki Sakuma[1], Takafumi Hasegawa[2], Ichiro Kawahata[3], Kohji Fukunaga[3], Atsushi Takeda[1,4]*

1 Department of Neurology, National Hospital Organization Sendai Nishitaga Hospital, Sendai, Japan, 2 Department of Neurology, Tohoku University Graduate School of Medicine, Sendai, Japan, 3 Department of Pharmacology, Tohoku University Graduate School of Pharmaceutical Sciences, Sendai, Japan, 4 Department of Cognitive and Motor Aging, Tohoku University Graduate School of Medicine, Sendai, Japan

* takeda.atsushi.nc@mail.hosp.go.jp

**Data Availability Statement:** All relevant data are within the paper and its Supporting information files.

**Funding:** The funding for this study was provided by grants-in-aid for Scientific Research from the

## Abstract

### Background

In recent years, there has been increasing evidence that several lipid metabolism abnormalities play an important role in the pathogenesis of neurodegenerative diseases. However, it is still unclear which lipid metabolism abnormalities play the most important role in neurodegenerative diseases. Plasma lipid metabolomics (lipidomics) has been shown to be an unbiased method that can be used to explore lipid metabolism abnormalities in neurodegenerative diseases. Plasma lipidomics in neurodegenerative diseases has been performed only in idiopathic Parkinson's disease (IPD) and Alzheimer's disease (AD), and comprehensive studies are needed to clarify the pathogenesis.

### Methods

In this study, we investigated plasma lipids using lipidomics in individuals with neurodegenerative diseases and healthy controls (CNs). Plasma lipidomics was evaluated by liquid chromatography-tandem mass spectrometry (LC–MS/MS) in those with IPD, dementia with Lewy bodies (DLB), multiple system atrophy (MSA), AD, and progressive supranuclear palsy (PSP) and CNs.

### Results

The results showed that (1) plasma sphingosine-1-phosphate (S1P) was significantly lower in all neurodegenerative disease groups (IPD, DLB, MSA, AD, and PSP) than in the CN group. (2) Plasma monohexylceramide (MonCer) and lactosylceramide (LacCer) were significantly higher in all neurodegenerative disease groups (IPD, DLB, MSA, AD, and PSP) than in the CN group. (3) Plasma MonCer levels were significantly positively correlated with plasma LacCer levels in all enrolled groups.

### Conclusion

S1P, Glucosylceramide (GlcCer), the main component of MonCer, and LacCer are sphingolipids that are biosynthesized from ceramide. Recent studies have suggested that elevated

Project of Translational and Clinical Research Core Centers from the Japan Agency for Medical Research and Development (AMED) (JP17dm0107071 and JP18dm0107071 to KF and AT). This work was supported by grants-in-aid from the Research Committee of CNS Degenerative Diseases, Research on Policy Planning and Evaluation for Rare and Intractable Diseases, Health, Labor and Welfare Sciences Research Grants, the Ministry of Health, Labor and Welfare, Japan.

**Competing interests:** The authors declare that the research was conducted in the absence of any commercial or financial relationships that could be construed as a potential conflict of interest.

**Abbreviations:** AD, Alzheimer's disease; C1P, ceramide-1-phosphate; CNs, controls; CSF, cerebrospinal fluid; DLB, dementia with Lewy bodies; GalCer, galactosylceramide; GCIs, glial cytoplasmic inclusions; GlcCer, glucosylceramide; IPD, Idiopathic Parkinson's disease; IS, Internal standards; LacCer, lactosylceramide; LBs, Lewy bodies; LC–MS/MS, liquid chromatography-tandem mass spectrometry; lipidomics, lipid metabolomics; LPA, lysophosphatidic acid; LPC, lysophosphatidylcholine; LPE, lysophosphatidylethanolamine; LPG, lysophosphatidylglycerol; LPI, lysophosphatidylinositol; LPS, lysophosphatidylserine; MMSE, Mini-Mental State Examination; MonCer, monohexylceramide; MSA, multiple system atrophy; NHO, National Hospital Organization; PAF, platelet-activating factor; PSP, progressive supranuclear palsy; S1P, sphingosine-1-phosphate; SG1P, sphinganine-1-phosphate.

GlcCer and decreased S1P levels in neurons are related to neuronal cell death and that elevated LacCer levels induce neurodegeneration by neuroinflammation. In the present study, we found decreased plasma S1P levels and elevated plasma MonCer and LacCer levels in those with neurodegenerative diseases, which is a new finding indicating the importance of abnormal sphingolipid metabolism in neurodegeneration.

# Introduction

The incidence of idiopathic Parkinson's disease (IPD) is reported to be 8–18 per 1000 person-years [1] and that of dementia with Lewy bodies (DLB) is 0.5–1.6 per 1000 person-years [2], which makes both of them common neurodegenerative diseases. Lewy body diseases, such as DLB and IPD, are characterized by the presence of cytoplasmic protein aggregates known as Lewy bodies (LBs) [3]. The main component of LBs is α-synuclein, which is abundant in neurons, including synaptic vesicles in presynaptic terminals, and is a protein aggregate that has been converted to a β-sheet fibril structure [4]. Multiple system atrophy (MSA) is an adult-onset neurodegenerative disease that is clinically characterized by poor levodopa-responsive parkinsonism, cerebellar dysfunction, and autonomic failure [5]. The histopathology of MSA is characterized by the presence of protein aggregates known as glial cytoplasmic inclusions (GCIs). Similar to LBs, GCIs are largely composed of aggregates of α-synuclein [6, 7]. Therefore, LB diseases and MSA are classified as neurodegenerative diseases named synucleinopathies, which are characterized by prominent intracellular α-synuclein aggregation [8].

Alzheimer's disease (AD) is the most common neurodegenerative disease, currently affecting approximately 40 million people worldwide [9]. In contrast to synucleinopathies, the pathological features of AD require the presence of extracellular β amyloid-positive senile plaques and phosphorylated tau-positive neurofibrillary tangles in neurons [10]. Progressive supranuclear palsy (PSP) is a neurodegenerative disease characterized by vertical supranuclear gaze palsy, postural instability and falls in the early stages of the disease [11]. The pathology of PSP is characterized by tau-positive aggregates with a characteristic 4-repeat tau in the microtubule-binding domain in neurons [12]. Therefore, AD and PSP have been classified as neurodegenerative diseases named tauopathies, which are characterized by prominent tau aggregation in neurons [13–15]. AD have been also classified as neurodegenerative diseases named amyloidopathies, which are characterized by prominent extracellular β amyloid aggregation [16].

Lipids are biomolecules that are soluble in nonpolar organic solvents, usually insoluble in water, and are known primarily for their metabolic role in energy storage [17]. Lipids are also major components of cell membranes and play an important role in cellular metabolism as components of lipid rafts, protein anchors, and signaling and transport molecules. There are eight distinct classes of lipids classified as fatty acyl, glycerolipids, glycerophospholipids, sphingolipids, sterols, prenols, saccharolipids, and polyketides [17]. Recently, abnormalities in cerebrospinal fluid (CSF) lipid metabolism have been reported in IPD and AD [18, 19]. CSF examination in neurodegenerative diseases is less costly than neuroimaging and more directly reflects the metabolic state and pathophysiology of the central nervous system than other body fluids, making it an important test for understanding pathophysiology. However, CSF testing is a rather invasive approach, and there is a need to develop more noninvasive methods of fluid collection (e.g., blood sampling) to evaluate the pathogenesis of neurodegenerative diseases.

In recent years, plasma metabolomics has attracted much attention as a method to search for metabolic abnormalities in an unbiased manner and one that reflects the pathophysiology

in vivo [20]. Lipid metabolites have various characteristics, such as molecular weight, polarity, and ionization state. For accurate analysis, it has been necessary to develop new tools that can detect a large number of lipid metabolites with high resolution. LC–MS/MS can detect a large number of lipid metabolites with high resolution and has attracted attention as a tool for lipid metabolomics (lipidomics) research. Plasma lipidomics in neurodegenerative diseases has been evaluated only in IPD and AD [21, 22], and comprehensive analysis is needed to clarify the pathogenesis. In the present study, we used plasma lipidomics to examine whether abnormalities in plasma lipid metabolism were observed in IPD, DLB, MSA, AD, and PSP.

## Materials and methods

### Clinical information of the participants in this study

All participants were recruited at National Hospital Organization (NHO) Sendai Nishitaga Hospital and examined by board-certified neurologists. Cohort A, cohort B, and cohort C were recruited from October 2017 to September 2021. Patients with IPD, probable DLB, probable AD, probable MSA, and probable PSP according to the established clinical diagnostic criteria for each disease were included [3, 5, 23–25]. All enrolled patients had late onset (>45 years of age), and no patients had a family history. All IPD patients were treated with L-dopa or other antiparkinsonian drugs, and motor symptoms were under good control. In cohort A, we enrolled 30 patients with IPD and 28 controls (CNs) (Table 1). The 30 IPD patients included 21 females and 9 males; the age of the IPD patients ranged from 58 to 75 years, with a mean of 67.2 years. The 28 CNs included 14 females and 14 males; the age of the CNs ranged from 57 to 73 years, with a mean of 65 years. In cohort B, 28 DLB patients, 13 AD patients, and

**Table 1. Demographics and clinical characteristics of the analyzed plasma samples in Cohorts A, B and C.**

| | cohort A | | | | | | |
|---|---|---|---|---|---|---|---|
| | CN | PD | p value (CN vs. PD) | | | | |
| number | 28 | 30 | | | | | |
| male, %/female, % | 14 (50)/14 (50) | 9 (30)/21 (70) | 0.1197 | | | | |
| age, y, mean±SD | 65.0±5.3 | 67.2±5.1 | 0.1095 | | | | |
| MMSE, mean±SD | 26.9±2.0 | 26.8±3.6 | 0.3532 | | | | |
| disease duration, y, mean±SD | | 9.2±6.1 | | | | | |
| | cohort B | | | | | | |
| | CN | DLB | p value (CN vs. DLB) | AD | p value (CN vs. AD) | | |
| number | 15 | 28 | | 13 | | | |
| male, %/female, % | 11 (73)/4 (27) | 11 (39)/17 (61) | **0.0333** | 2 (15)/11 (85) | **0.0022** | | |
| age, y, mean±SD | 66.8±5.2 | 83.3±6.2 | **<0.0001** | 83.6±4.5 | **<0.0001** | | |
| MMSE, mean±SD | 26.7±2.1 | 24.2±6.7 | **0.0217** | 19.6±4.9 | **<0.0001** | | |
| disease duration, y, mean±SD | | 3.4±3.4 | | 1.5±2.1 | | | |
| | cohort C | | | | | | |
| | CN | PD | p value (CN vs. PD) | PSP | p value (CN vs. PSP) | MSA | p value (CN vs. MSA) |
| number | 6 | 28 | | 16 | | 13 | |
| male, %/female, % | 4 (67)/2 (33) | 9 (32)/19 (68) | 0.1143 | 9 (56)/7 (44) | 0.6581 | 6 (46)/7 (54) | 0.4052 |
| age, y, mean±SD | 71.5±1.2 | 75.2±5.9 | 0.1077 | 74.8±6.8 | 0.086 | 69.6±11.9 | 0.9299 |
| MMSE, mean±SD | 28.8±1 | 24.0±3.8 | **0.0014** | 22.5±4.5 | **0.0079** | 26.2±1.3 | 0.0572 |
| disease duration, y, mean±SD | | 6.4±6.6 | | 3.6±2.2 | | 2.8±1.7 | |

Abbreviations: CNs, controls; PD, Parkinson's disease; DLB, dementia with Lewy bodies; Alzheimer's disease (AD); SD, standard deviation; MMSE: Mini-Mental State Examination; PSP, progressive supranuclear palsy; MSA, multiple system atrophy

15 CNs were enrolled (Table 1). The 28 DLB patients included 17 females and 11 males; the age of the DLB patients ranged from 72 to 95 years, with a mean of 83.3 years. The 13 AD patients included 11 females and 2 males; the age of the AD patients ranged from 73 to 88 years, with a mean of 83.6 years. The 15 CNs included 4 females and 11 males; the age of the CNs ranged from 55 to 73 years, with a mean of 66.8 years. In cohort C, 28 PD patients, 13 MSA patients, 16 PSP patients, and 6 CNs were enrolled (Table 1). The 28 IPD patients included 19 females and 9 males; the age of the IPD patients ranged from 60 to 85 years, with a mean of 75.2 years. The 13 MSA patients included 7 females and 6 males; the age of the MSA patients ranged from 50 to 92 years, with a mean of 69.6 years. The 16 PSP patients included 7 females and 9 males; the age of the PSP patients ranged from 60 to 84 years, with an average of 74.8 years; 6 CNs included 2 females and 4 males; the age of the CNs ranged from 70 to 73 years, with an average of 71.5 years.

In this study, duration of illness refers to the time since the onset of motor symptoms in the IPD, MSA, and PSP patients and the onset of cognitive impairment in the DLB and AD patients. The Mini-Mental State Examination (MMSE) was used as a global cognitive function test. All CNs, all DLB patients, all AD patients, 24 out of 30 IPD patients in cohort A, 23 out of 28 IPD patients in cohort C, 10 out of 13 MSA patients, and 16 out of 18 PSP patients completed the MMSE.

This study was approved by the ethics committee of our institution and followed the Helsinki Declaration on International Clinical Research Involving Human Beings. Written informed consent for this study was obtained from all subjects.

## Sample collection

Sample collection was performed from October 2017 to September 2021. Plasma was extracted as previously described [26]. Each 500 μl plasma aliquot was stored in a -80˚C freezer until use. Briefly, fasting blood was collected in Na-EDTA and centrifuged at room temperature for 10 minutes to extract plasma. The extracted plasma was collected in screw-cap microtubes (Sarstedt AG, Nümbrecht, Germany) between 10 am and 12 am and stored at -80˚C until the time of metabolomic analysis.

## Metabolite extraction

Metabolite extraction and metabolomic analysis were conducted at Human Metabolome Technologies (HMT) (HMT, Tsuruoka, Yamagata, Japan). Briefly, 100 μL of plasma was mixed with 300 μL of 0.1% formic acid in methanol containing internal standards and centrifuged at 9,100 ×$g$ and 4˚C for 10 minutes. Then, 250 μL of the supernatant was mixed with 550 μL of 0.1% formic acid and loaded onto an SPE column (MonoSpinC18, 5010–2170, GL Sciences Inc., Tokyo, Japan). The analytes on the SPE column were purified with 0.1% formic acid and 0.1% formic acid in 25% methanol and eluted with 200 μL of 0.1% formic acid in methanol. The elution was then used for LC–MS/MS analysis at HMT. The average recovery of sphingolipids extracted with the SPE column is 88% (range 68% to 99.9%).

**Metabolomic analysis.** Metabolomic analysis was conducted by the *Mediator Scan* package of HMT by using LC–MS/MS. Based on metabolomic analysis, 324 metabolites, including fatty acids, acylcarnitines, oxylipins, lysophospholipids, platelet-activating factors, glycosphingolipids, sphinganines, sphingosines, and steroids, were evaluated in all enrolled neurodegenerative disease groups and the CN group. Briefly, LC–MS/MS analysis was carried out by using an Agilent 1260 Infinity II and Agilent 1290 Infinity II High Speed Pump equipped with AB Sciex QTRAP 5500 (AB Sciex Pte. Ltd., Framingham, MA, USA). The multiple reaction

monitoring (MRM) mode of the mass spectrometer was used to detect signals of each metabolite according to the HMT metabolite database. MRM ion chromatograms were extracted by using Multi Quant automatic integration software (AB Sciex) to obtain peak area information. Target metabolites are divided into categories (fatty acids, acylcarnitines, oxylipins, lysophospholipids, platelet-activating factors, glycosphingolipids, sphinganines, sphingosines, and steroids) according to their physical properties, and the recovery rate is corrected using the corresponding IS (Internal standards). Based on these reports, these IS were selected [27–29]. The recovery rate of analytes during extraction ranged from 68% to 129%, with a mean of 96%. IS coefficient of variation ranged from 4.4 to 9.7%, with a mean of 6.7%. The peak area of each metabolite was then normalized based on IS level and sample volume for relative quantification. The normalized each metabolite was represented as relative area and used as the quantitative value based on previous reports [30, 31].

## Simoa™ assay

Plasma samples stored at -80˚C were thawed and centrifuged at 10,000 x g for 5 minutes. Samples were diluted in advance with the Sample Diluent provided with Assay Kit and applied to the plate. The assay was performed one sample at a time. Simoa™ p-Tau181 Advantage Kit (Quanterix, #103377, Billerica, MA, USA) were used to measure plasma p-Tau181. Measurements were performed according to the instructions for kit.

## Statistical evaluation

All plasma metabolites are expressed as the median (interquartile range). Differences between the groups were examined for statistical significance using one-tailed Welch's t test in the lipidomic analysis data. Differences between the groups were examined for statistical significance using Wilcoxon tests and chi-square tests for the demographic data. Data were analyzed using the computer software system JMP13 (SAS Institute, Tokyo, Japan).

## Results

### Plasma sphingosine-1-phosphate (S1P) levels in neurodegenerative diseases

Plasma S1P levels were compared between the CN group and the IPD, DLB, MSA, AD and PSP groups. Statistical significance was examined using one-tailed Welch's t tests. Plasma S1P d16.1 levels were significantly ($p < 0.0001$) lower in the IPD group of cohort A (N = 30) versus the control group (N = 28) (Fig 1A). Plasma S1P d16.1 levels were significantly ($p < 0.0001$) lower in the DLB group (N = 28) versus the control group (N = 15) (Fig 1B) and significantly ($p < 0.0001$) lower in the AD group (N = 13) versus the control group (N = 15) (Fig 1B). Plasma S1P d16.1 levels were significantly ($p < 0.01$) lower in the IPD group of cohort C (N = 28) versus the control group (N = 6) (Fig 1C), significantly ($p < 0.01$) lower in the MSA group (N = 13) versus the control group (N = 6) (Fig 1C), and significantly ($p < 0.001$) lower in the PSP group (N = 16) versus the control group (N = 6) (Fig 1C). Plasma S1P d18.1 levels were significantly ($p < 0.05$) lower in the IPD group of cohort A (N = 30) versus the control group (N = 28) (Fig 1D). Plasma S1P d18.1 levels were significantly ($p < 0.01$) lower in the DLB group (N = 28) versus the control group (N = 15) (Fig 1E) and significantly ($p < 0.05$) lower in the AD group (N = 13) versus the control group (N = 15) (Fig 1E). Plasma S1P d18.1 levels were significantly ($p < 0.05$) lower in the IPD group of cohort C (N = 28) versus the control group (N = 6) (Fig 1F), significantly ($p < 0.05$) lower in the MSA group (N = 13) versus the control group (N = 6) (Fig 1F), and significantly ($p < 0.05$) lower in the PSP group (N = 16) versus the control group (N = 6) (Fig 1F). These results indicated that plasma S1P

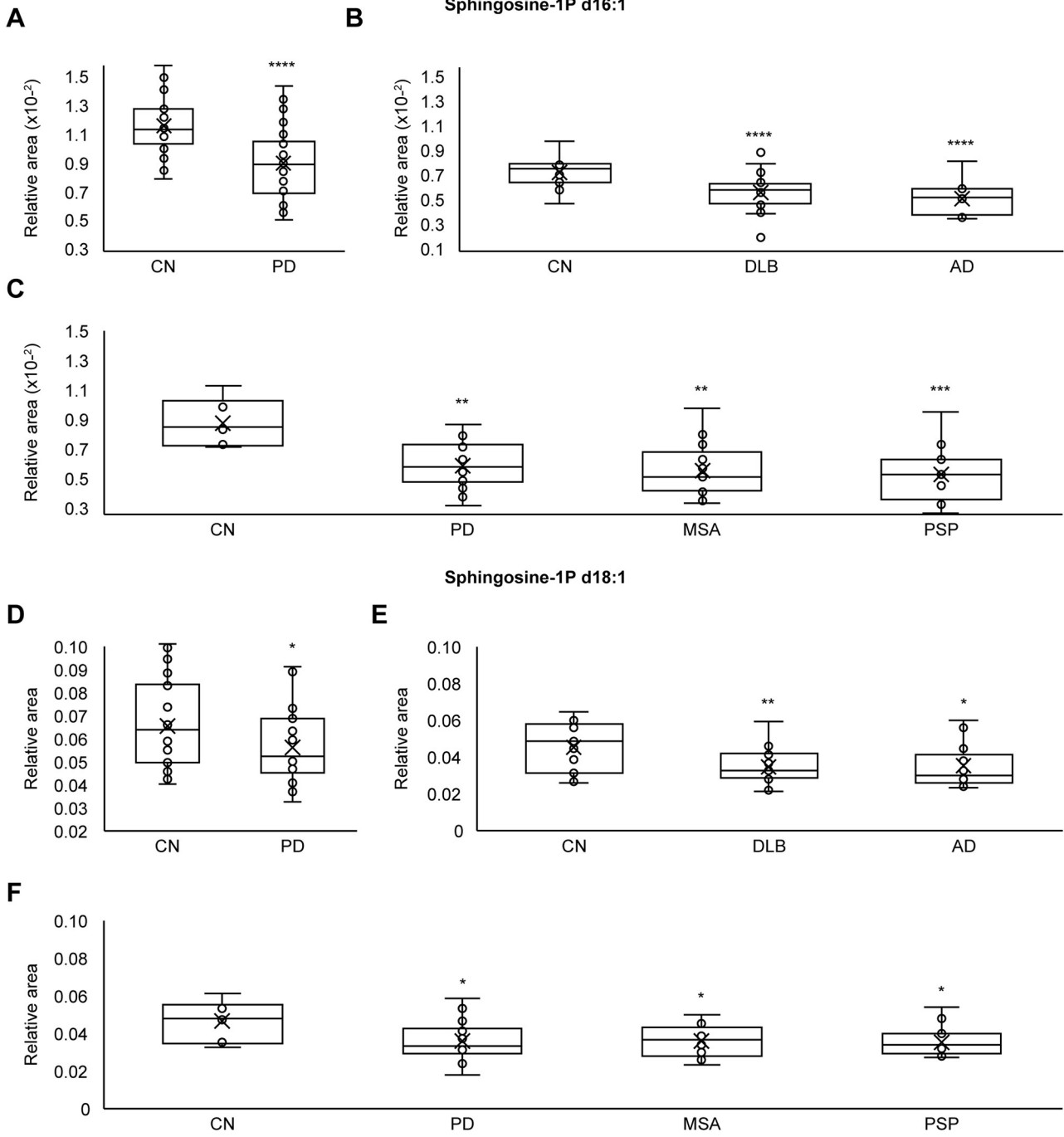

**Fig 1. Plasma S1P levels in neurodegenerative diseases.** (A) Plasma S1P d16.1 levels were significantly lower in the IPD group of cohort A ($p < 0.0001$) than in the CN group. (B) Plasma S1P d16.1 levels were significantly lower in the DLB group ($p < 0.0001$) and AD group ($p < 0.0001$) than in the CN group. (C) Plasma S1P d16.1 levels were significantly lower in the IPD group of cohort C ($p < 0.01$), MSA group ($p < 0.01$) and PSP group ($p < 0.001$) than in the CN group. (D) Plasma S1P d18.1 levels were significantly lower in the IPD group of cohort A ($p < 0.05$) than in the CN group. (E) Plasma S1P d18.1 levels were significantly lower in the DLB group ($p < 0.01$) and AD group ($p < 0.05$) than in the CN group. (F) Plasma S1P d18.1 levels were significantly lower in the IPD group of cohort C ($p < 0.05$), MSA group ($p < 0.05$) and PSP group ($p < 0.05$) than in the CN group. Statistical significance was examined using one-tailed Welch's t tests ($P < 0.05$). Circles indicate the data points between the lower and upper whiskers, and x indicates the average marker in a box/whisker diagram.

levels were significantly lower in all neurodegenerative disease groups (IPD, DLB, MSA, AD, and PSP) than in the CN group.

## Plasma monohexylceramide (MonCer) levels in neurodegenerative diseases

Total plasma MonCer d18:1 levels were compared between the CN group and the IPD, DLB, MSA, AD, and PSP groups. Total plasma MonCer d18:1 levels were measured by summing levels of 13 types of MonCer d18:1: MonCer (d18:1/14:0), MonCer (d18:1/16:0), MonCer (d18:1/16:1), MonCer (d18:1/18:0), MonCer (d18:1/18:1), MonCer (d18:1/20:0), MonCer (d18:1/20:1), MonCer (d18:1/22:0), MonCer (d18:1/22:1), MonCer (d18:1/22:2), MonCer (d18:1/24:0), MonCer (d18:1/24:1) and MonCer (d18:1/24:2). Statistical significance was examined using one-tailed Welch's t tests. Total plasma MonCer d18:1 levels were significantly ($p < 0.01$) higher in the IPD group of cohort A (N = 30) versus the control group (N = 28) (Fig 2A). Total plasma MonCer d18:1 levels were significantly ($p < 0.01$) higher in the DLB group (N = 28) versus the control group (N = 15) (Fig 2B) and significantly ($p < 0.001$) higher in the AD group (N = 13) versus the control group (N = 15) (Fig 2B). Total plasma MonCer d18:1 levels were significantly ($p < 0.01$) higher in the IPD group of cohort C (N = 28) versus the control group (N = 6) (Fig 2C), significantly ($p < 0.05$) higher in the MSA group (N = 13) versus the control group (N = 6) (Fig 2C), and significantly ($p < 0.01$) higher in the PSP group (N = 16) versus the control group (N = 6) (Fig 2C). These results indicated that plasma MonCer levels were significantly higher in all neurodegenerative disease groups (IPD, DLB, MSA, AD, and PSP) than in the CN group.

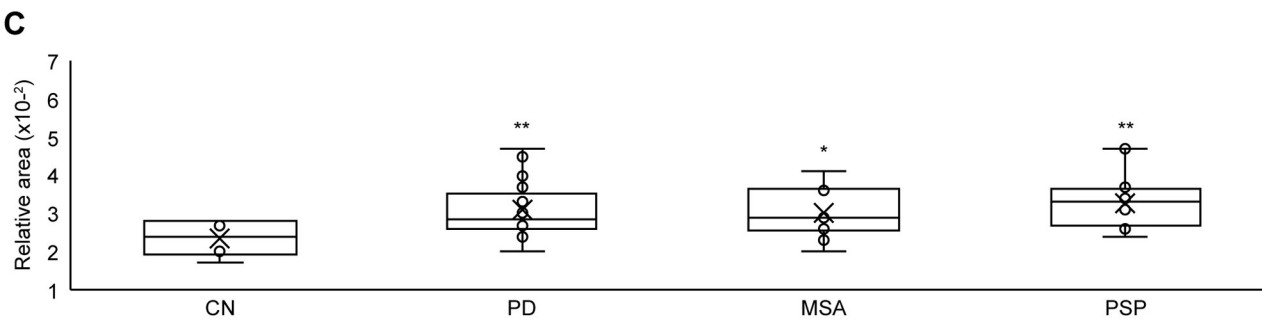

**Fig 2. Plasma MonCer levels in neurodegenerative diseases.** (A) Plasma MonCer d18:1 levels were significantly higher in the IPD group of cohort A ($p < 0.01$) than in the CN group. (B) Plasma MonCer d18:1 levels were significantly higher in the DLB group ($p < 0.01$) and AD group ($p < 0.001$) than in the CN group. (C) Plasma MonCer d18:1 levels were significantly higher in the IPD group of cohort C ($p < 0.01$), MSA group ($p < 0.05$) and PSP group ($p < 0.01$) than in the CN group. Statistical significance was examined using one-tailed Welch's t tests (P < 0.05). Circles indicate the data points between the lower and upper whiskers, and x indicates the average marker in a box/whisker diagram.

**Table 2. Plasma all MonCer levels in neurodegenerative diseases.**

| cohort A | | | cohort B | | | cohort B | | |
| --- | --- | --- | --- | --- | --- | --- | --- | --- |
| PD vs CN | | | DLB vs CN | | | AD vs CN | | |
| | ratio | p value | | ratio | p value | | ratio | p value |
| MonCer (d18:1/14:0) | 1.1 | 0.0975 | MonCer (d18:1/14:0) | 1.3 | **0.0012** | MonCer (d18:1/14:0) | 1.2 | **0.0443** |
| MonCer (d18:1/16:0) | 1.2 | **0.0151** | MonCer (d18:1/16:0) | 1.2 | **0.0013** | MonCer (d18:1/16:0) | 1.3 | **0.0023** |
| MonCer (d18:1/16:1) | 1 | 0.5408 | MonCer (d18:1/16:1) | 1.1 | 0.2052 | MonCer (d18:1/16:1) | 1 | 0.4006 |
| MonCer (d18:1/18:0) | 1.3 | **0.0105** | MonCer (d18:1/18:0) | 1.2 | **0.0181** | MonCer (d18:1/18:0) | 1.3 | **0.0008** |
| MonCer (d18:1/18:1) | 1.4 | **0.0018** | MonCer (d18:1/18:1) | 1.3 | **0.0042** | MonCer (d18:1/18:1) | 1.3 | **0.0106** |
| MonCer (d18:1/20:0) | 1.2 | **0.0053** | MonCer (d18:1/20:0) | 1.1 | 0.0890 | MonCer (d18:1/20:0) | 1.2 | **0.0298** |
| MonCer (d18:1/20:1) | 1.5 | **0.0010** | MonCer (d18:1/20:1) | 1.4 | **0.0066** | MonCer (d18:1/20:1) | 1.4 | **0.0047** |
| MonCer (d18:1/22:0) | 1.1 | 0.2312 | MonCer (d18:1/22:0) | 1 | 0.5454 | MonCer (d18:1/22:0) | 1.1 | 0.2288 |
| MonCer (d18:1/22:1) | 1.4 | **0.0092** | MonCer (d18:1/22:1) | 1.5 | **0.0025** | MonCer (d18:1/22:1) | 1.5 | **0.0121** |
| MonCer (d18:1/22:2) | 1.3 | **0.0097** | MonCer (d18:1/22:2) | 1.4 | **0.0034** | MonCer (d18:1/22:2) | 1.5 | **0.0039** |
| MonCer (d18:1/24:0) | 1.1 | 0.2478 | MonCer (d18:1/24:0) | 1 | 0.5483 | MonCer (d18:1/24:0) | 0.9 | 0.6499 |
| MonCer (d18:1/24:1) | 1.3 | **0.0016** | MonCer (d18:1/24:1) | 1.3 | **0.0143** | MonCer (d18:1/24:1) | 1.3 | **0.0244** |
| MonCer (d18:1/24:2) | 1.2 | **0.0176** | MonCer (d18:1/24:2) | 1.2 | **0.0170** | MonCer (d18:1/24:2) | 1.4 | **0.0313** |
| cohort C | | | cohort C | | | cohort C | | |
| PD vs CN | | | PSP vs CN | | | MSA vs CN | | |
| | ratio | **p value** | | ratio | **p value** | | ratio | **p value** |
| MonCer (d18:1/14:0) | 1.3 | **0.0826** | MonCer (d18:1/14:0) | 1.3 | **0.0709** | MonCer (d18:1/14:0) | 1.2 | **0.1345** |
| MonCer (d18:1/16:0) | 1.4 | **0.0029** | MonCer (d18:1/16:0) | 1.4 | **0.0035** | MonCer (d18:1/16:0) | 1.4 | **0.0099** |
| MonCer (d18:1/16:1) | 1.2 | **0.3156** | MonCer (d18:1/16:1) | 1.5 | **0.1082** | MonCer (d18:1/16:1) | 1.3 | **0.2442** |
| MonCer (d18:1/18:0) | 1.5 | **0.0018** | MonCer (d18:1/18:0) | 1.5 | **0.0030** | MonCer (d18:1/18:0) | 1.4 | **0.0074** |
| MonCer (d18:1/18:1) | 1.3 | **0.0018** | MonCer (d18:1/18:1) | 1.5 | **0.0006** | MonCer (d18:1/18:1) | 1.2 | **0.0910** |
| MonCer (d18:1/20:0) | 1.3 | **0.0267** | MonCer (d18:1/20:0) | 1.4 | **0.0100** | MonCer (d18:1/20:0) | 1.3 | **0.0430** |
| MonCer (d18:1/20:1) | 1.3 | **0.0182** | MonCer (d18:1/20:1) | 1.5 | **0.0020** | MonCer (d18:1/20:1) | 1.3 | **0.0172** |
| MonCer (d18:1/22:0) | 0.8 | **0.7890** | MonCer (d18:1/22:0) | 1.1 | **0.1771** | MonCer (d18:1/22:0) | 0.9 | **0.7486** |
| MonCer (d18:1/22:1) | 1 | **0.5938** | MonCer (d18:1/22:1) | 1.4 | **0.0729** | MonCer (d18:1/22:1) | 1.2 | **0.2142** |
| MonCer (d18:1/22:2) | 1.5 | **0.0009** | MonCer (d18:1/22:2) | 1.6 | **0.0013** | MonCer (d18:1/22:2) | 1.6 | **0.0189** |
| MonCer (d18:1/24:0) | 0.4 | **<0.0001** | MonCer (d18:1/24:0) | 1.1 | **0.2573** | MonCer (d18:1/24:0) | 0.7 | **0.0158** |
| MonCer (d18:1/24:1) | 0.9 | **0.2863** | MonCer (d18:1/24:1) | 1.4 | **0.0136** | MonCer (d18:1/24:1) | 1.1 | **0.3389** |
| MonCer (d18:1/24:2) | 1.2 | **0.0725** | MonCer (d18:1/24:2) | 1.5 | **0.0145** | MonCer (d18:1/24:2) | 1.3 | **0.0697** |

Statistical methods: The metabolite level ratio of IPD, DLB, MSA, AD, or PSP to CNs. Statistical significance was examined using one-tailed Welch's t tests ($P < 0.05$).

We compared MonCer (d18:1/14:0), MonCer (d18:1/16:0), MonCer (d18:1/16:1), MonCer (d18:1/18:0), MonCer (d18:1/18:1), MonCer (d18:1/20:0), MonCer (d18:1/20:1), MonCer (d18:1/22:0), MonCer (d18:1/22:1), MonCer (d18:1/22:2), MonCer (d18:1/24:0), MonCer (d18:1/24:1), and MonCer (d18:1/24:2) between the CN group and the IPD, DLB, MSA, AD, or PSP groups (Table 2). The chi-square test was used to examine the association between lipid abnormalities and chain length in MonCer d18:1. No statistically significant difference was found between lipid abnormalities and chain length ($P = 0.5522$) in all enrolled groups.

## Plasma lactosylceramide (LacCer) levels in neurodegenerative diseases

Total plasma LacCer d18:1 levels were compared between the CN group and the IPD, DLB, MSA, AD, and PSP groups. Total plasma LacCer d18:1 levels were measured by summing the levels of 13 types of LacCer d18:1: LacCer (d18:1/14:0), LacCer (d18:1/16:0), LacCer (d18:1/16:1), LacCer (d18:1/18:0), LacCer (d18:1/18:1), LacCer (d18:1/20:0), LacCer (d18:1/20:1),

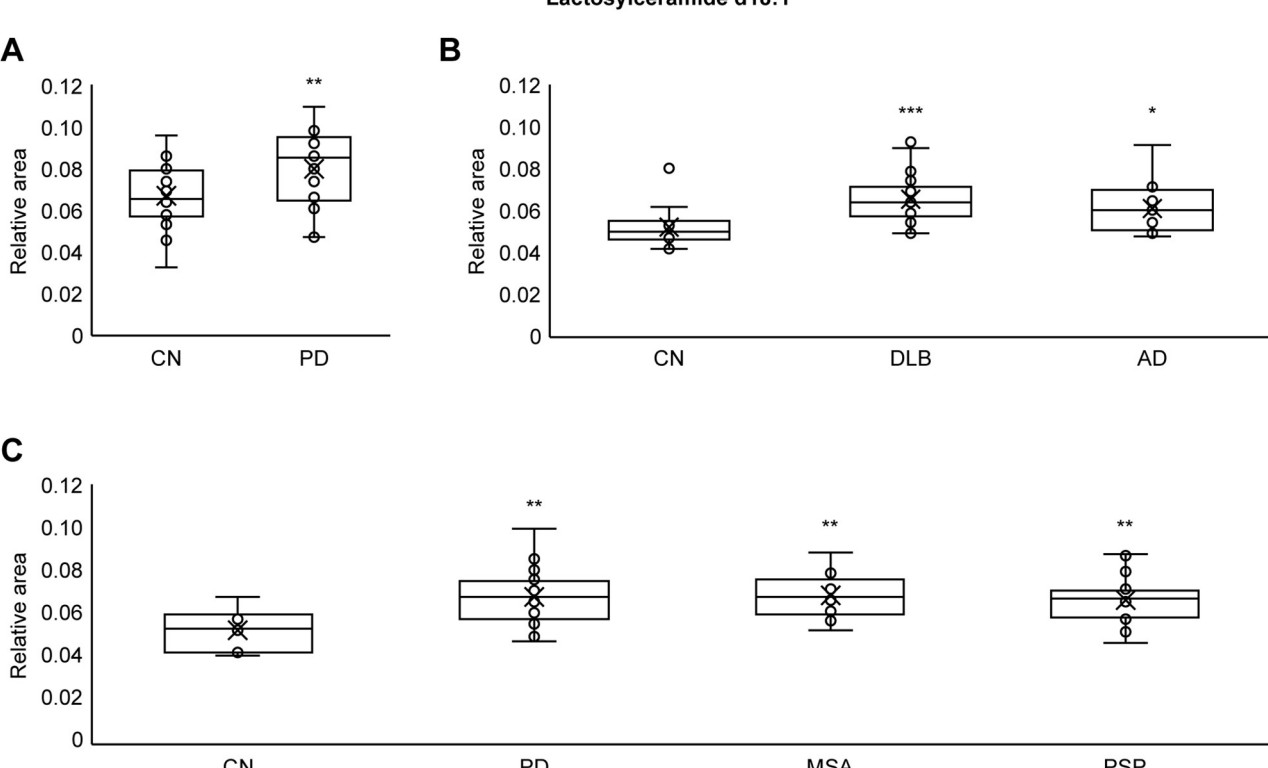

**Fig 3. Plasma LacCer levels in neurodegenerative diseases.** (A) Plasma LacCer d18:1 levels were significantly higher in the IPD group of cohort A (p < 0.01) than in the CN group. (B) Plasma LacCer d18:1 levels were significantly higher in the DLB group (p < 0.001) and AD group (p < 0.05) than in the CN group. (C) Plasma LacCer d18:1 levels were significantly higher in the IPD group of cohort C (p < 0.01), MSA group (p < 0.01) and PSP group (p < 0.01) than in the CN group. Statistical significance was examined using one-tailed Welch's t tests (P < 0.05). Circles indicate the data points between the lower and upper whiskers, and x indicates the average marker in a box/whisker diagram.

LacCer (d18:1/22:0), LacCer (d18:1/22:1), LacCer (d18:1/22:2), LacCer (d18:1/24:0), LacCer (d18:1/24:1) and LacCer (d18:1/24:2). Statistical significance was examined using one-tailed Welch's t tests. Total plasma LacCer d18:1 levels were significantly (p < 0.01) higher in the IPD group of cohort A (N = 30) versus the control group (N = 28) (Fig 3A). Total plasma LacCer d18:1 levels were significantly (p < 0.001) higher in the DLB group (N = 28) versus the control group (N = 15) (Fig 3B) and significantly (p < 0.05) higher in the AD group (N = 13) versus the control group (N = 15) (Fig 3B). Total plasma LacCer d18:1 levels were significantly (p < 0.01) higher in the IPD group of cohort C (N = 28) versus the control group (N = 6) (Fig 3C), significantly (p < 0.01) higher in the MSA group (N = 13) versus the control group (N = 6) (Fig 3C), and significantly (p < 0.01) higher in the PSP group (N = 16) versus the control group (N = 6) (Fig 3C). These results indicated that plasma LacCer levels were significantly higher in all neurodegenerative disease groups (IPD, DLB, MSA, AD, and PSP) than in the CN group.

We compared LacCer (d18:1/14:0), LacCer (d18:1/16:0), LacCer (d18:1/16:1), LacCer (d18:1/18:0), LacCer (d18:1/18:1), LacCer (d18:1/20:0), LacCer (d18:1/20:1), LacCer (d18:1/22:0), LacCer (d18:1/22:1), LacCer (d18:1/22:2), LacCer (d18:1/24:0), LacCer (d18:1/24:1), and LacCer (d18:1/24:2) between the CN group and the IPD, DLB, MSA, AD, or PSP groups (Table 3). The chi-square test was used to examine the association between lipid abnormalities and chain length in LacCers d18:1. No statistically significant difference was found between lipid abnormalities and chain length (P = 0.5522) in all enrolled groups.

**Table 3. Plasma All LacCer levels in neurodegenerative diseases.**

| cohort A | | | cohort B | | | cohort B | | |
|---|---|---|---|---|---|---|---|---|
| PD vs CN | | | DLB vs CN | | | AD vs CN | | |
| | ratio | p value | | ratio | p value | | ratio | p value |
| LacCer (d18:1/14:0) | 1.2 | **0.0035** | LacCer (d18:1/14:0) | 1.5 | **<0.0001** | LacCer (d18:1/14:0) | 1.2 | **0.0297** |
| LacCer (d18:1/16:0) | 1.2 | **0.0026** | LacCer (d18:1/16:0) | 1.2 | **0.0006** | LacCer (d18:1/16:0) | 1.2 | **0.0284** |
| LacCer (d18:1/16:1) | 1.2 | **0.0161** | LacCer (d18:1/16:1) | 1.3 | **0.0002** | LacCer (d18:1/16:1) | 1.2 | 0.0655 |
| LacCer (d18:1/18:0) | 1.1 | 0.1188 | LacCer (d18:1/18:0) | 1.2 | **0.0110** | LacCer (d18:1/18:0) | 1.2 | **0.0340** |
| LacCer (d18:1/18:1) | 1.3 | **0.0024** | LacCer (d18:1/18:1) | 1.2 | **0.0278** | LacCer (d18:1/18:1) | 1.2 | 0.0850 |
| LacCer (d18:1/20:0) | 1 | 0.3364 | LacCer (d18:1/20:0) | 1.1 | 0.1147 | LacCer (d18:1/20:0) | 1.2 | 0.1116 |
| LacCer (d18:1/20:1) | 1.4 | **0.0032** | LacCer (d18:1/20:1) | 1.4 | **0.0009** | LacCer (d18:1/20:1) | 1.3 | **0.0179** |
| LacCer (d18:1/22:0) | 1 | 0.5292 | LacCer (d18:1/22:0) | 1.1 | 0.2660 | LacCer (d18:1/22:0) | 1.1 | 0.2379 |
| LacCer (d18:1/22:1) | 1.3 | **0.0095** | LacCer (d18:1/22:1) | 1.5 | **<0.0001** | LacCer (d18:1/22:1) | 1.4 | **0.0167** |
| LacCer (d18:1/22:2) | 1.3 | **0.0018** | LacCer (d18:1/22:2) | 1.3 | **0.0062** | LacCer (d18:1/22:2) | 1.3 | **0.0268** |
| LacCer (d18:1/24:0) | 1 | 0.4800 | LacCer (d18:1/24:0) | 1.1 | 0.3120 | LacCer (d18:1/24:0) | 1 | 0.6157 |
| LacCer (d18:1/24:1) | 1.4 | **0.0003** | LacCer (d18:1/24:1) | 1.3 | **0.0082** | LacCer (d18:1/24:1) | 1.2 | 0.1190 |
| LacCer (d18:1/24:2) | 1.3 | **0.0021** | LacCer (d18:1/24:2) | 1.3 | **0.0098** | LacCer (d18:1/24:2) | 1.1 | 0.1727 |
| cohort C | | | cohort C | | | cohort C | | |
| PD vs CN | | | PSP vs CN | | | MSA vs CN | | |
| | ratio | **p value** | | ratio | **p value** | | ratio | p value |
| LacCer (d18:1/14:0) | 1.4 | **0.0252** | LacCer (d18:1/14:0) | 1.3 | **0.0320** | LacCer (d18:1/14:0) | 1.4 | 0.0194 |
| LacCer (d18:1/16:0) | 1.3 | **0.0063** | LacCer (d18:1/16:0) | 1.4 | **0.0019** | LacCer (d18:1/16:0) | 1.3 | 0.0059 |
| LacCer (d18:1/16:1) | 1.2 | **0.0157** | LacCer (d18:1/16:1) | 1.5 | **0.0178** | LacCer (d18:1/16:1) | 1.2 | 0.0454 |
| LacCer (d18:1/18:0) | 1.2 | **0.0953** | LacCer (d18:1/18:0) | 1.5 | **0.2933** | LacCer (d18:1/18:0) | 1.3 | 0.0392 |
| LacCer (d18:1/18:1) | 1.2 | **0.1763** | LacCer (d18:1/18:1) | 1.5 | **0.1542** | LacCer (d18:1/18:1) | 1 | 0.4351 |
| LacCer (d18:1/20:0) | 1.1 | **0.2281** | LacCer (d18:1/20:0) | 1.4 | **0.3886** | LacCer (d18:1/20:0) | 1.2 | 0.1381 |
| LacCer (d18:1/20:1) | 1.2 | **0.0902** | LacCer (d18:1/20:1) | 1.5 | **0.2285** | LacCer (d18:1/20:1) | 1.2 | 0.1321 |
| LacCer (d18:1/22:0) | 0.8 | **0.1630** | LacCer (d18:1/22:0) | 1.1 | **0.6517** | LacCer (d18:1/22:0) | 0.9 | 0.2980 |
| LacCer (d18:1/22:1) | 1.1 | **0.3525** | LacCer (d18:1/22:1) | 1.4 | **0.2833** | LacCer (d18:1/22:1) | 1.1 | 0.2590 |
| LacCer (d18:1/22:2) | 1.3 | **0.0925** | LacCer (d18:1/22:2) | 1.6 | **0.1774** | LacCer (d18:1/22:2) | 1.2 | 0.1781 |
| LacCer (d18:1/24:0) | 0.4 | **0.0031** | LacCer (d18:1/24:0) | 1.1 | **0.7710** | LacCer (d18:1/24:0) | 0.7 | 0.0248 |
| LacCer (d18:1/24:1) | 0.9 | **0.6587** | LacCer (d18:1/24:1) | 1.4 | **0.3145** | LacCer (d18:1/24:1) | 1 | 0.4450 |
| LacCer (d18:1/24:2) | 1.2 | **0.2138** | LacCer (d18:1/24:2) | 1.5 | **0.2520** | LacCer (d18:1/24:2) | 1.2 | 0.2394 |

Statistical methods: The metabolite level ratio of IPD, DLB, MSA, AD, or PSP to CNs. Statistical significance was examined using one-tailed Welch's t tests (P < 0.05).

## Correlation between total plasma MonCer levels and total plasma LacCer levels

Pearson Correlation Coefficient was used to correlate total plasma MonCer d18:1 levels and total plasma LacCer d18:1 levels in all enrolled groups. Total plasma MonCer d18:1 levels were significantly positively correlated with total plasma LacCer d18:1 levels (r = 0.5802, p < 0.0001) (Fig 4) in all enrolled groups. These results suggest that an increase in plasma MonCer may be directly related to an increase in LacCer in all enrolled groups.

## Correlation between plasma p-tau levels and plasma S1P levels, total plasma MonCer levels or total plasma LacCer levels

To investigate the association between AD-associated protein and sphingolipids, Pearson Correlation Coefficient was used to correlate plasma p-tau levels and plasma S1P d16.1 levels,

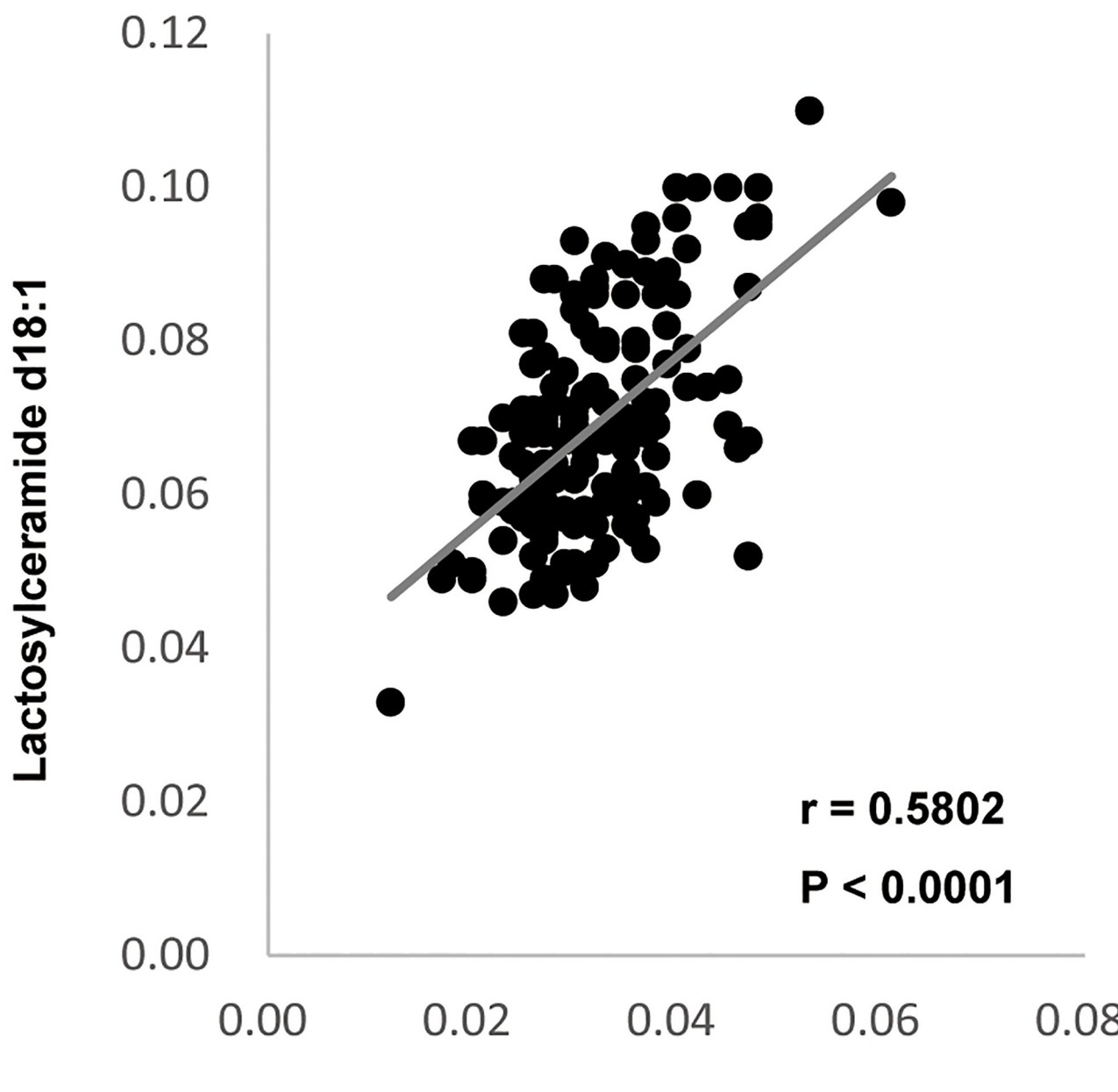

**Fig 4. Correlation between total plasma MonCer levels and total plasma LacCer levels.** (A) Total plasma MonCer d18:1 levels were significantly positively correlated with total plasma LacCer d18:1 levels (r = 0.5802, p < 0.0001) in all enrolled groups.

plasma S1P d18.1 levels, total plasma MonCer d18.1 levels or total plasma LacCer d18.1 levels in all enrolled groups. Correlation between plasma p-tau levels and plasma S1P d16.1 levels (p = 0.509), plasma S1P d18.1 levels (p = 0.468), plasma MonCer d18.1 levels (p = 0.767), or plasma LacCer d18.1 levels (p = 0.999) showed no correlation.

## Plasma other lipid metabolite levels in neurodegenerative diseases

Plasma other lipid metabolite (other sphingolipids, sphinganines, gangliosides, free fatty acids, acylcarnitnes, lysophospholipids, platelet-activating factor, acylethanolamine, thyroid hormone, cholic acids, and steroids) levels were compared between the CN group and the IPD, DLB, MSA, AD and PSP groups. Oxylipins were not statistically analyzed because it is considered unsuitable for statistical analysis due to the large number of undetectable samples. Statistical significance was examined using one-tailed Welch's t tests. Plasma ceramide-1-phosphate (C1P) levels were significantly higher in the PD, DLB, and AD groups versus the control group (S1 Table). Plasma GM3 ganglioside and GD3 ganglioside levels were significantly higher in all neurodegenerative disease groups (IPD, DLB, MSA, AD, and PSP) versus the control group (S1 Table). Plasma lysophosphatidic acid, lysophosphatidylcholine, lysophosphatidylethanolamine, lysophosphatidylglycerol, lysophosphatidylserine levels were lower in DLB group versus the control group (S1 Table). Plasma cortisone levels were significantly higher in the PD, MSA and PSP groups versus the control group (S2 Table).

## Discussion

### Plasma sphingolipid abnormalities in neurodegenerative diseases

Recessive mutations in the GBA1 (glucocerebrosidase) gene cause Gaucher disease. Heterozygous GBA1 mutation carriers exhibit much greater incidence of PD than the general population [32, 33]. Likewise, mutations in the NPC1 (NPC intracellular cholesterol transporter 1) and SMPD1 (sphingomyelin phosphodiesterase 1) genes, which cause Niemann-Pick disease, have been shown to be risk genes for IPD [34, 35]. One of the phospholipase A2 members, PLA2G6 or iPLA2-VIA/iPLA2β, has been isolated as the gene responsible for an autosomal recessive form of PD linked to the PARK14 locus [36]. Compared to the most common e3 isoform, the e4 isoform of ApoE (ApoE4) is the strongest genetic risk factor for late-onset AD [37]. β amyloid accumulation in NPC1 (NPC intracellular cholesterol transporter 1) gene, which cause Niemann-Pick type C, mutant cells and NPC mouse brain suggests the association between cholesterol metabolism and AD [38]. As described, several lipid-related genes have been reported as risk genes or causative genes in PD and AD. In addition, various lipid abnormalities have been reported in IPD and AD, such as fatty acids, glycerolipids, glycerophospholipids, sphingolipids, sterols, and lipoproteins [17, 39]. However, it is still unclear which lipid metabolism abnormalities play the most important role in neurodegenerative diseases. Plasma lipidomics is an unbiased method and can find important lipids in neurodegenerative diseases. For this reason, plasma lipidomics was performed in neurodegenerative diseases in this study. In this study, we found that plasma S1P levels were significantly lower and plasma MonCer and LacCer levels were significantly higher in all neurodegenerative disease groups (IPD, DLB, MSA, AD, and PSP) than in the CN group by plasma lipidomics.

Glucosylceramide (GlcCer) and galactosylceramide (GalCer) are isomers, and MonCer is the sum of both compounds. Although it is difficult to completely separate plasma GalCer and plasma GlcCer from plasma MonCer in present method, it has been shown that the majority of plasma MonCer is composed of plasma GlcCer [40]. S1P, GlcCer, and LacCer mentioned above are sphingolipids biosynthesized from ceramide (Fig 5). GCS is GlcCer synthase, BGTase6 is LacCer synthase, and SPHK is S1P synthase. These indicate that increased GlcCer and LacCer are caused by increased function of GCS and BGTase6, respectively, and decreased S1P is caused by a relative loss of function of SPHK.

Ceramide is hydrolyzed to sphingosine, which is further phosphorylated by sphingosine kinase to S1P (Fig 5). S1P is a sphingolipid that regulates stress tolerance, proliferation and

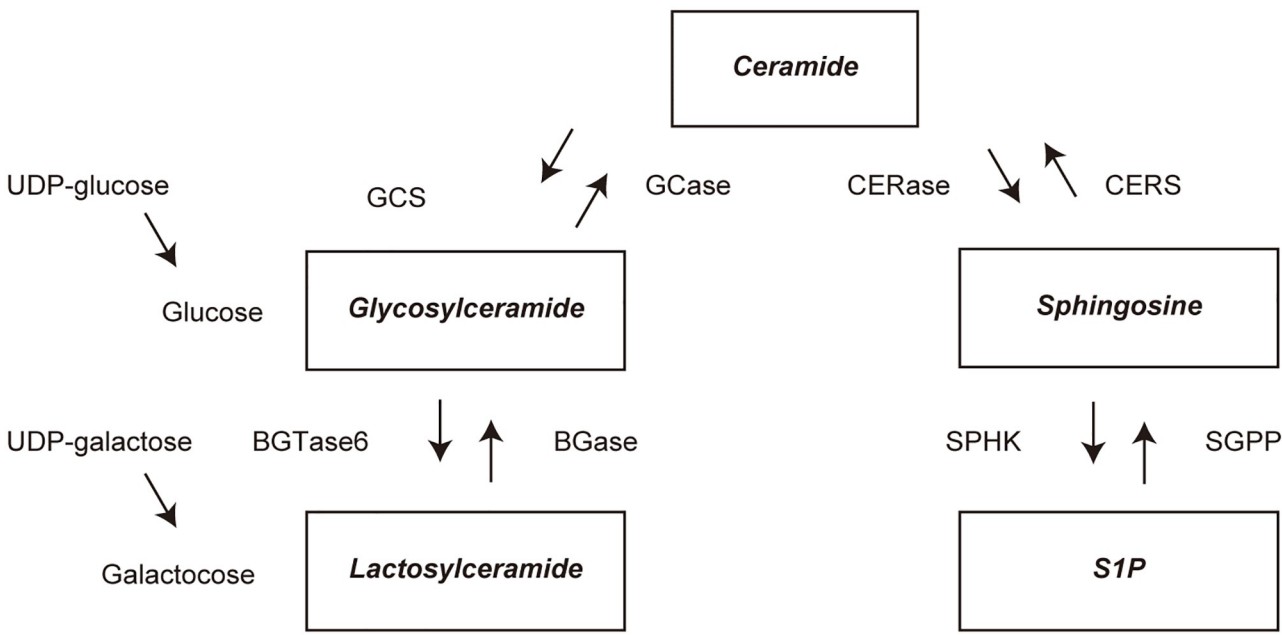

**Fig 5. Ceramide, sphingosine and glycosphingolipid metabolism.** Products are indicated in bold and italics. Abbreviations: S1P, sphingosine-1-phosphate; GlcCer, glucosylceramide; GCase, glucocerebrosidase; GCS, GlcCer synthase; CERase, ceramidase; CERS, ceramide synthase; SPHK, sphingosine kinase; SGPP, S1P phosphatase; BGTase6, beta-1,4-galactosyltransferase 6; BGase, beta-galactosidase.

differentiation of neuronal cells and is a neuroprotective factor involved in the suppression of neuronal cell death [41, 42]. It has been reported that S1P concentrations in CSF are significantly decreased in AD [43], and S1P concentrations in plasma are significantly decreased in vascular dementia and AD [44]. However, there has been no comprehensive analysis of plasma S1P levels in those with neurodegenerative diseases such as synucleinopathies, amyloidopathies and tauopathies. Therefore, we analyzed plasma S1P levels in individuals with neurodegenerative diseases using lipidomics in this study. We found that plasma S1P levels in all neurodegenerative disease groups (IPD, DLB, MSA, AD, and PSP) were significantly lower than those in the CN group. The finding of lower plasma S1P levels in those with all neurodegenerative disease groups analyzed (IPD, DLB, MSA, AD, and PSP) is a novel finding revealed in this study, suggesting that abnormalities in plasma S1P metabolism are common in synucleinopathies, amyloidopathies and tauopathies.

In animal models of the synucleinopathies PD and MSA administration of FTY720, an S1P agonist, has been shown to ameliorate neurodegeneration and behavioral dysfunction associated with mitochondrial dysfunction via S1P receptors [45, 46]. α-synuclein binds to lipid rafts, where it negatively regulates S1P receptor signaling [47]. S1P levels were decreased with increasing Braak stage in AD, and this was most pronounced in brain regions most affected by AD pathology [48]. In an animal model of AD in which Aβ42 peptide was injected locally into the bilateral hippocampus, administration of the S1P agonist FTY720 reduced hippocampal neuronal damage and learning and memory impairment [49]. Furthermore, in an animal model of AD using rat hippocampal slices, administration of SEW2871, an S1P agonist, was shown to suppress the expression of phosphorylated tau protein [50]. These findings suggest that S1P may act as a neuroprotective factor against aggregate formation and neuronal cell death not only in PD but also in AD. In other words, the decrease in plasma S1P levels in synucleinopathies and amyloidopathies may reflect a decrease in neuroprotection.

GlcCer is generated by glucosylceramide synthase (GCS), which transfers glucose from UDP-glucose to ceramide (Fig 5). GlcCer is a glycosphingolipid that regulates lysosomal function in general. Plasma GlcCer (a MonCer) levels have been shown to be significantly elevated in PD, autopsy-confirmed DLB, and autopsy-confirmed AD groups [51, 52]. However, there has been no comprehensive analysis of plasma GlcCer (a MonCer) levels in neurodegenerative diseases such as synucleinopathies and tauopathies. Therefore, we analyzed plasma GlcCer (a MonCer) in those with neurodegenerative diseases using lipidomics in this study. We found that the plasma GlcCer (a MonCer) levels were significantly higher in all neurodegenerative disease groups (IPD, DLB, MSA, AD, and PSP) than in the CN group. The elevated plasma GlcCer (a MonCer) levels in individuals with IPD, probable DLB, and probable AD in this study were in good accordance with the results of previous studies [51, 52]. There have been no reports of abnormal plasma GlcCer (a MonCer) levels in MSA and PSP. In this study, we found elevated plasma GlcCer (a MonCer) levels in individuals not only with LB diseases or AD but also with MSA or PSP, suggesting that abnormalities in plasma GlcCer (a MonCer) metabolism are also commonly observed in synucleinopathies, amyloidopathies and tauopathies.

GBA1 is a major causative gene for Gaucher disease. Recently, GBA1 mutations have been reported to be an important risk factor for LB diseases such as IPD and DLB [53, 54]. The GBA1 mutation reduces the activity of the lysosomal lipid metabolizing enzyme glucocerebrosidase (GCase), which catalyzes the hydrolysis of the glycosphingolipid GlcCer into ceramide and glucose, resulting in increased intracellular GlcCer levels [55]. Interestingly, elevated plasma GlcCer levels have recently been reported in both non-GBA1 mutation carriers and GBA1 mutation carriers with IPD [52, 56]. In GBA1 mutation carriers with IPD, decreased GCase activity promoted elevated intracellular GlcCer levels and increased α-synuclein aggregation [57], and this aggregation resulted in a loss of lysosomal activity and neuronal death [58–60]. In the pathological brain tissue of IPD patients without GBA1 mutations, GCase activity was also reported to be decreased [61]. This suggested that increased plasma GlcCer levels are observed in IPD with or without the GBA1 mutation and that increased intraneuronal GlcCer levels may be involved in aggregation formation and neuronal cell death. Presenilin mutation, one of the familial AD genes, is strongly involved in Aβ42 aggregation, the main component of senile plaques, and a previous report showed that presenilin deficiencies resulted in increased GlcCer synthase levels [62]. Furthermore, it has been shown that GlcCer levels were increased in the brain tissue of those with idiopathic AD [63]. This suggested that elevated GlcCer levels in the brain are also present in AD and are related to disease pathology.

LacCer is generated by LacCer synthase (β-1,4 galactosyltransferase), which transfers galactose from UDP-galactose to GlcCer (Fig 5). Plasma LacCer levels were significantly elevated in the non-GBA1 mutation carrier IPD group compared to the CN group [52]. We found that the plasma LacCer levels were significantly higher in all neurodegenerative disease groups (IPD, DLB, MSA, AD, and PSP) than in the CN group. In this study, elevated plasma LacCer levels in those with IPD were in good accordance with the results of a previous study [52]. In this study, we found elevated plasma LacCer levels not only in those with IPD but also those with DLB, MSA, AD or PSP, suggesting that abnormalities in plasma LacCer metabolism are also commonly observed in synucleinopathies, amyloidopathies and tauopathies.

LacCer is a glycosphingolipid, which is an important component of "lipid rafts," serving as a conduit to transduce external stimuli [64]. As biologically active sphingolipids, LacCer plays diverse roles in inflammation, cell proliferation, migration/infiltration, adhesion, angiogenesis apoptosis, autophagy, and mitochondrial dysfunction [64]. LacCer generally induces neurodegeneration in the central nervous system by activating astrocytes that regulate neuroinflammation [65]. Thus, elevated plasma LacCer levels may reflect neuroinflammation in the central nervous system.

In this study, we found that plasma GM3 and GD3 ganglioside levels were significantly higher in the neurodegenerative disease groups than in the CN group. Gangliosides are lipids classified as sphingolipids. GM3 ganglioside is the starting material for gangliosides, which are biosynthesized by the binding of sialic acid to LacCer [66, 67]. Previously, plasma GM3 ganglioside levels have been shown to be elevated in PD [68]. The elevated plasma GM3 ganglioside levels in individuals with IPD in this study were in good accordance with the results of previous study. GD3 ganglioside is the gangliosides, which are biosynthesized by the binding of sialic acid to GM3 ganglioside [66, 67]. GM3 and GD3 gangliosides are components of lipid rafts and are implicated in cell death [69, 70]. Abnormalities in lipid rafts are also considered to be one of the major causes of neurodegenerative diseases [71]. Homozygous knockout mice for B4galnt1, a ganglioside synthase, have been shown to exhibit PD-like motor deficits and cause dopaminergic neuron degeneration [72]. Taken together, these results suggest that elevated plasma GM3 and GD3 gangliosides may reflect abnormal lipid rafts in neurodegenerative diseases. In this study, we found that plasma C1P levels were significantly higher in the IPD, DLB, and AD groups than in the CN group. C1P is classified as a sphingolipid, a lipid mainly involved in cell survival and inflammation [73, 74]. Neuroinflammation is also considered to be a one of the major causes in PD, DLB and AD [75–77]. Therefore, elevated C1P may reflect neuroinflammation in these diseases.

## Limitations of this study

There are several limitations in this study. First, Analysis the major causative genes or risk genes of PD during lipidomics were not evaluated. GBA1 mutations were not evaluated in all enrolled IPD patients. Based on the GBA1 genotype and clinical analysis, it has been reported that GBA1 mutation is the most common genetic risk factor for IPD patients, accounting for as many as 7% of all IPD patients in multicenter analyses [32, 33]. On the other hand, only approximately 3% of Asian IPD patients with no apparent family history of parkinsonism are GBA1 mutation carriers [78]. IPD in GBA1 mutation carriers generally has an early onset [53]. However, there was no apparent family history of parkinsonism or dementia in all enrolled IPD patients, with a later mean age of onset in the enrolled IPD patients that was 67.2 years in cohort A and 65.2 years in cohort C. Elevated plasma GlcCer levels have recently been reported in GBA1 mutation carriers of IPD. Elevated plasma GlcCer levels have also been reported in non-GBA1 mutation carriers of IPD. These indicate that elevated plasma GlcCer is found in IPD with or without GBA mutation. Taken together, it is not plausible that a GBA1 mutation did not significantly affect elevated plasma GlcCer (a MonCer) levels in the IPD patients in this study. In addition, in this study LRRK2 and SNCA mutations, the major causative genes of PD, were not evaluated in all enrolled IPD patients. Analysis the major causative genes or risk genes of PD during lipidomics need to be performed in future studies. Second, this study is a small cases and cross-sectional study that could not account for multiple comparisons for several analytes detected in plasma. Future additional cases and longitudinal studies need to be performed. Third, a major limitation of this study is that the patients were not pathologically diagnosed. Fourth, we were not able to include other dementia diseases, such as frontotemporal dementia. Fifth, cohort B was not an age-matched study. In DLB and AD, correlation analysis between age and plasma S1P d16:1 levels, plasma S1P d18:1 levels, plasma MonCer d18:1 levels, or plasma LacCer d18:1 levels showed no correlation (S3 Table). Thus, changes in plasma S1P d16:1 levels, plasma S1P d18:1 levels, plasma MonCer d18:1 levels or plasma LacCer d18:1 levels were inferred to be disease-induced changes in AD or DLB. Sixth, in this study the protein levels of the enzymes involved in sphingolipid pathways were not evaluated in all enrolled patients. The protein levels of the enzymes involved in sphingolipid pathways need to be performed in future studies. Seventh, relative area was used in this study as the

quantitative value for each metabolite based on previous reports [30, 31]. Lipidomics has the variability of metabolite values in each study. For this reason, each metabolite should be normalized based on the IS level and sample volume. The normalized each metabolite was represented as relative area and used as the quantitative value. Eighth, the increase and decrease in CSF sphingolipids and blood sphingolipids have coincided [30, 31, 57, 58] in previous reports. On the other hand, one report even identified different findings in serum versus CSF [53, 79]. These reports are indirect and sphingolipids need to be confirmed in CSF or brain for further validation.

## Summary of the results

Using plasma lipidomics analysis, we identified decreased plasma S1P levels and increased plasma GlcCer (a MonCer) and LacCer levels in individuals with neurodegenerative diseases. These abnormalities in plasma sphingolipids might be closely related to aggregate formation, neuronal cell death and neuroinflammation. Our results provide new insights into the involvement of sphingolipids in neurodegenerative diseases.

## Supporting information

**S1 Table. Plasma other sphingolipids, sphinganines, gangliosides, free fatty acids, acylcarnitnes, lysophospholipids levels in neurodegenerative diseases.** Statistical methods: The metabolite level ratio of IPD, DLB, MSA, AD, or PSP to CNs. Statistical significance was examined using one-tailed Welch's t tests (P < 0.05). Abbreviations: ceramide-1-phosphate (C1P), sphinganine-1-phosphate (SG1P), lysophosphatidic acid (LPA), lysophosphatidylcholine (LPC), lysophosphatidylethanolamine (LPE), lysophosphatidylglycerol (LPG), lysophosphatidylinositol (LPI), lysophosphatidylserine (LPS).
(DOCX)

**S2 Table. Plasma platelet-activating factor, acylethanolamine, thyroid hormone, cholic acids, steroids levels in neurodegenerative diseases.** Statistical methods: The metabolite level ratio of IPD, DLB, MSA, AD, or PSP to CNs. Statistical significance was examined using one-tailed Welch's t tests (P < 0.05). Abbreviations: platelet-activating factor (PAF).
(DOCX)

**S3 Table. Correlation analysis between age and plasma S1P d16:1 levels, plasma S1P d18:1 levels, plasma MonCer d18:1 levels, or plasma LacCer d18:1 levels in DLB and AD.** Pearson Correlation Coefficient was used to correlate between age and plasma S1P d16:1 levels, plasma S1P d18:1 levels, plasma MonCer d18:1 levels, or plasma LacCer d18:1 levels (P < 0.05).
(DOCX)

## Acknowledgments

We gratefully acknowledge Shiryu Takemura for technical support.

## Author Contributions

**Conceptualization:** Hideki Oizumi, Takafumi Hasegawa, Atsushi Takeda.

**Data curation:** Hideki Oizumi, Yoko Sugimura, Tomoko Totsune, Iori Kawasaki, Saki Ohshiro, Toru Baba, Teiko Kimpara, Hiroaki Sakuma, Ichiro Kawahata.

**Formal analysis:** Hideki Oizumi, Ichiro Kawahata.

**Funding acquisition:** Kohji Fukunaga, Atsushi Takeda.

**Investigation:** Hideki Oizumi, Ichiro Kawahata.

**Writing – original draft:** Hideki Oizumi.

**Writing – review & editing:** Atsushi Takeda.

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
