## [Decision Letter · Decision Letter 0]

27 Jun 2022

PONE-D-22-15953Plasma sphingolipid abnormalities in neurodegenerative diseasesPLOS ONE

Dear Dr. Takeda,

Thank you for submitting your manuscript to PLOS ONE. After careful consideration, we feel that it has merit but does not fully meet PLOS ONE’s publication criteria as it currently stands. Therefore, we invite you to submit a revised version of the manuscript that addresses the points raised during the review process.

Comments of reviewer #1 Please address these two issues:1. Information on the separation of GalCer from GlcCer is necessary to understand to which compound are referred the results.

2. Quantitative data are necessary to understand the yield of extraction and the correct determination and comparison within samples. Comments of reviewer #2 Comments 1: Do you have information on fluid biomarkers such as plasma Ab42/Ab40 ratio and ptau? Comment 2: Can you address the question as to how changes in plasma sphingolipids relate to brain and CSF changes? Comment 3: you must address the issue relating to the different ages (are changes age-related or AD-related?) Comment 4: you must address this concern. Comment 5:  please answer these two questions comment 6: address the issue regarding adding statements in the limitations section Note: Given that age is a significant risk factor in Alzheimer’s disease, the age differences should be factored into the statistical analyses.

We look forward to receiving your revised manuscript.

Kind regards,

Stephan N. Witt, Ph.D.

Academic Editor

PLOS ONE

Journal Requirements:

" ext-link-type="uri" xlink:type="simple">https://journals.plos.org/plosone/s/file?id=ba62/PLOSOne_formatting_sample_title_authors_affiliations.pdf"

2. Please amend the manuscript submission data (via Edit Submission) to include authors Yoko Sugimura, Iori Kawasaki, Saki Ohshiro, Toru Baba, Teiko Kimpara, and Hiroaki Sakuma.

Additional Editor Comments:

Major issue:

lines 190-213 (Results section on S1P): The authors repeat in words what is depicted in the figures. This is far too repetitive. What is the purpose of the figures, if all of the findings are given in words?

Example: " Plasma

S1P d16.1 levels were significantly lower in the IPD group of cohort A (plasma S1P d16.1 levels:

30 IPD vs. 28 CNs; 0.0091 (0.0103-0.00713) vs. 0.0115 (0.013-0.0103); t = 4.34, p 0.0001) (Fig 1A),

Suggested way to rephrase the above sentence:

Plasma S1P d16.1 levels were significantly (p,0.0001) lower in the IPD group of cohort A (N=

30) versus the control group (N=28) (Fig 1A)..."

Bottom line: the entire RESULTS section of the manuscript should be simplified by eliminating this redundancy of saying in words what is given in the figures.

Reviewers' comments:

Reviewer's Responses to Questions

**Comments to the Author**

1. Is the manuscript technically sound, and do the data support the conclusions?

Reviewer #1: Partly

Reviewer #2: Yes

2. Has the statistical analysis been performed appropriately and rigorously? 

Reviewer #1: Yes

Reviewer #2: No

3. Have the authors made all data underlying the findings in their manuscript fully available?

Reviewer #1: Yes

Reviewer #2: No

4. Is the manuscript presented in an intelligible fashion and written in standard English?

Reviewer #1: Yes

Reviewer #2: Yes

5. Review Comments to the Author

Reviewer #1: This manuscript is interesting, but very preliminary. Only a few sphingolipids on the total species present in plasma were analyzed. Of course the analysis of plasma lipids from GBA1 associated neurodegenerative disease patients would be necessary, but I believe that more important would be the knowledge on the others, not considered sphingolipids.

Information on the separation of GalCer from GlcCer is necessary to understand to which compound are referred the results.

Quantitative data are necessary to understand the yield of extraction and the correct determination and comparison within samples.

Reviewer #2: General Comment to authors

The authors examine plasma sphingolipid in several neurodegenerative diseases and show striking changes in sphingosine-1-phosphate and GluCer and LacCer species. However, the small sample sizes and the significant age differences in cohort B are concerns, given that age is a significant risk factor in Alzheimer’s disease.

Specific Comments to Authors

1) Interestingly, sphingolipids change in all neurodegenerative diseases examined. How do the changes in S1P, GluCer, and LacCer relate to known changes in fluid biomarkers such as plasma A�42 and A�40 ratio, and plasma ptau levels?. It would be necessary also to measure these fluid biomarkers and not merely rely on MMSE, given the small sample sizes.

2) How do the plasma changes in sphingolipids relate to brain and CSF changes?

3) The age of the CN population (cohort B) is significantly lower than that of the DLB and AD populations. Since age is a significant risk factor for AD, it is unclear whether the changes are AD-related or age-related. Additional analyses with age as a variable may help tease out whether these data are age-related or pathological.

4) What is the recovery of sphingolipids extracted with the SPE column (page 11)? What internal standards were used, and how quantitative were these standards?

5) Is there a correlation/link between the decrease in S1P and the increase in GlcCer and LacCer in plasma? What enzyme pathways may cause the decrease in S1P and the increase in GluCer and LacCer?

6) The limitations (page 22) should include that this is a small discovery/cross sectional study that can not account for multiple comparisons for several analytes detected in plasma.

6. PLOS authors have the option to publish the peer review history of their article (what does this mean?). If published, this will include your full peer review and any attached files.

Reviewer #1: No

Reviewer #2: No

---

## [Author Response · Author response to Decision Letter 0]

22 Sep 2022

Additional Editor Comments:

Major issue:

lines 190-213 (Results section on S1P): The authors repeat in words what is depicted in the figures. This is far too repetitive. What is the purpose of the figures, if all of the findings are given in words?

Example: " Plasma

S1P d16.1 levels were significantly lower in the IPD group of cohort A (plasma S1P d16.1 levels:

30 IPD vs. 28 CNs; 0.0091 (0.0103-0.00713) vs. 0.0115 (0.013-0.0103); t = 4.34, p 0.0001) (Fig 1A),

Suggested way to rephrase the above sentence:

Plasma S1P d16.1 levels were significantly (p,0.0001) lower in the IPD group of cohort A (N=30) versus the control group (N=28) (Fig 1A)..."

Thank you for your suggestion. We agree with your opinion. We added sentences in Results section as you have indicated. 

Plasma S1P d16.1 levels were significantly (p 0.0001) lower in the IPD group of cohort A (N=30) versus the control group (N=28) (Fig 1A). Plasma S1P d16.1 levels were significantly (p 0.0001) lower in the DLB group (N=28) versus the control group (N=15) (Fig 1B) and significantly (p 0.0001) lower in the AD group (N=13) versus the control group (N=15) (Fig 1B). Plasma S1P d16.1 levels were significantly (p 0.01) lower in the IPD group of cohort C (N=28) versus the control group (N=6) (Fig 1C), significantly (p 0.01) lower in the MSA group (N=13) versus the control group (N=6) (Fig 1C), and significantly (p 0.001) lower in the PSP group (N=16) versus the control group (N=6) (Fig 1C). Plasma S1P d18.1 levels were significantly (p 0.05) lower in the IPD group of cohort A (N=30) versus the control group (N=28) (Fig 1D). Plasma S1P d18.1 levels were significantly (p 0.01) lower in the DLB group (N=28) versus the control group (N=15) (Fig 1E) and significantly (p 0.05) lower in the AD group (N=13) versus the control group (N=15) (Fig 1E). Plasma S1P d18.1 levels were significantly (p 0.05) lower in the IPD group of cohort C (N=28) versus the control group (N=6) (Fig 1F), significantly (p 0.05) lower in the MSA group (N=13) versus the control group (N=6) (Fig 1F), and significantly (p 0.05) lower in the PSP group (N=16) versus the control group (N=6) (Fig 1F). (p13-14, lines 204-219)

Total plasma MonCer d18:1 levels were significantly (p 0.01) higher in the IPD group of cohort A (N=30) versus the control group (N=28) (Fig 2A). Total plasma MonCer d18:1 levels were significantly (p 0.01) higher in the DLB group (N=28) versus the control group (N=15) (Fig 2B) and significantly (p 0.001) higher in the AD group (N=13) versus the control group (N=15) (Fig 2B). Total plasma MonCer d18:1 levels were significantly (p 0.01) higher in the IPD group of cohort C (N=28) versus the control group (N=6) (Fig 2C), significantly (p 0.05) higher in the MSA group (N=13) versus the control group (N=6) (Fig 2C), and significantly (p 0.01) higher in the PSP group (N=16) versus the control group (N=6) (Fig 2C). (p15, lines 243-251)

Total plasma LacCer d18:1 levels were significantly (p 0.01) higher in the IPD group of cohort A (N=30) versus the control group (N=28) (Fig 3A). Total plasma LacCer d18:1 levels were significantly (p 0.001) higher in the DLB group (N=28) versus the control group (N=15) (Fig 3B) and significantly (p 0.05) higher in the AD group (N=13) versus the control group (N=15) (Fig 3B). Total plasma LacCer d18:1 levels were significantly (p 0.01) higher in the IPD group of cohort C (N=28) versus the control group (N=6) (Fig 3C), significantly (p 0.01) higher in the MSA group (N=13) versus the control group (N=6) (Fig 3C), and significantly (p 0.01) higher in the PSP group (N=16) versus the control group (N=6) (Fig 3C). (p16-17, lines 270-278)

Reviewers' comments:

Reviewer's Responses to Questions

Reviewer #1: This manuscript is interesting, but very preliminary. Only a few sphingolipids on the total species present in plasma were analyzed. Of course the analysis of plasma lipids from GBA1 associated neurodegenerative disease patients would be necessary, but I believe that more important would be the knowledge on the others, not considered sphingolipids.

1)

Information on the separation of GalCer from GlcCer is necessary to understand to which compound are referred the results.

Thank you for your important remarks.

Unfortunately, it has been reported that the majority of plasma Monohexylceramides (MonCer) is composed of plasma GlcCer [1], but it is difficult to completely separate plasma GalCer from plasma GlcCer with present methods. Thank you for your advice. As you have indicated, plasma MonCer is the sum of plasma GalCer and plasma GlcCer, so we described plasma GlcCer as plasma MonCer. In addition, Mielke et al. also described plasma GlcCer as GlcCer (a MonCer) in their PROS ONE manuscript [2]. Thus, we also described plasma GlcCer as GlcCer (a MonCer) in reference to their PROS ONE manuscript. 

In response to your suggestion, we have added text to the Abstract, Results and Discussion sections, and have referred to plasma GlcCer as plasma MonCer or GlcCer (a MonCer) in this manuscript. 

 (2) Plasma monohexylceramide (MonCer) and lactosylceramide (LacCer) were significantly higher in all neurodegenerative disease groups (IPD, DLB, MSA, AD, and PSP) than in the CN group. (3) Plasma MonCer levels were significantly positively correlated with plasma LacCer levels in all enrolled groups. Conclusion: S1P, Glucosylceramide (GlcCer), the main component of MonCer, and LacCer are sphingolipids that are biosynthesized from ceramide. (p3, lines 48-52)

Plasma Monohexylceramide (MonCer) levels in neurodegenerative diseases

Total plasma MonCer d18:1 levels were compared between the CN group and the IPD, DLB, MSA, AD, and PSP groups. Total plasma MonCer d18:1 levels were measured by summing levels of 13 types of MonCer d18:1: MonCer (d18:1/14:0), MonCer (d18:1/16:0), MonCer (d18:1/16:1), MonCer (d18:1/18:0), MonCer (d18:1/18:1), MonCer (d18:1/20:0), MonCer (d18:1/20:1), MonCer (d18:1/22:0), MonCer (d18:1/22:1), MonCer (d18:1/22:2), MonCer (d18:1/24:0), MonCer (d18:1/24:1) and MonCer (d18:1/24:2). Statistical significance was examined using one-tailed Welch's t tests. Total plasma MonCer d18:1 levels were significantly (p 0.01) higher in the IPD group of cohort A (N=30) versus the control group (N=28) (Fig 2A). Total plasma MonCer d18:1 levels were significantly (p 0.01) higher in the DLB group (N=28) versus the control group (N=15) (Fig 2B) and significantly (p 0.001) higher in the AD group (N=13) versus the control group (N=15) (Fig 2B). Total plasma MonCer d18:1 levels were significantly (p 0.01) higher in the IPD group of cohort C (N=28) versus the control group (N=6) (Fig 2C), significantly (p 0.05) higher in the MSA group (N=13) versus the control group (N=6) (Fig 2C), and significantly (p 0.01) higher in the PSP group (N=16) versus the control group (N=6) (Fig 2C). These results indicated that plasma MonCer levels were significantly higher in all neurodegenerative disease groups (IPD, DLB, MSA, AD, and PSP) than in the CN group.

Fig 2. Plasma MonCer Levels in Neurodegenerative Diseases. (A) Plasma MonCer d18:1 levels were significantly higher in the IPD group of cohort A (p 0.01) than in the CN group. (B) Plasma MonCer d18:1 levels were significantly higher in the DLB group (p 0.01) and AD group (p 0.001) than in the CN group. (C) Plasma MonCer d18:1 levels were significantly higher in the IPD group of cohort C (p 0.01), MSA group (p 0.05) and PSP group (p 0.01) than in the CN group. (p15-16, lines 235-260)

Correlation between total plasma MonCer levels and total plasma LacCer levels.

Pearson Correlation Coefficient was used to correlate total plasma MonCer d18:1 levels and total plasma LacCer d18:1 levels in all enrolled groups. Total plasma MonCer d18:1 levels were significantly positively correlated with total plasma LacCer d18:1 levels (r = 0.5802, p 0.0001) (Fig 4) in all enrolled groups. These results suggest that an increase in plasma MonCer may be directly related to an increase in LacCer in all enrolled groups.

Fig 4. Correlation between total plasma MonCer levels and total plasma LacCer levels. (A) Total plasma MonCer d18:1 levels were significantly positively correlated with total plasma LacCer d18:1 levels (r = 0.5802, p 0.0001) in all enrolled groups.

(p17-18, lines 289-298)

In this study, we found that plasma S1P levels were significantly lower and plasma MonCer and LacCer levels were significantly higher in the neurodegenerative disease groups than in the CN group in a plasma lipidomics study. Glucosylceramide (GlcCer) and galactosylceramide (GalCer) are isomers, and MonCer is the sum of both compounds. Although it is difficult to completely separate plasma GalCer and plasma GlcCer from plasma MonCer in present method, it has been shown that the majority of plasma MonCer is composed of plasma GlcCer [1]. (p19, lines 312-317)

Plasma GlcCer (a MonCer) levels have been shown to be significantly elevated in PD, autopsy-confirmed DLB, and autopsy-confirmed AD groups [2,3]. However, there has been no comprehensive analysis of plasma GlcCer (a MonCer) levels in neurodegenerative diseases such as synucleinopathies and tauopathies. Therefore, we analyzed plasma GlcCer (a MonCer) in those with neurodegenerative diseases using lipidomics in this study. We found that the plasma GlcCer (a MonCer) levels were significantly higher in all neurodegenerative disease groups (IPD, DLB, MSA, AD, and PSP) than in the CN group. The elevated plasma GlcCer (a MonCer) levels in individuals with IPD, probable DLB, and probable AD in this study were in good accordance with the results of previous studies [2,3]. There have been no reports of abnormal plasma GlcCer (a MonCer) levels in MSA and PSP. In this study, we found elevated plasma GlcCer (a MonCer) levels in individuals not only with LB diseases or AD but also with MSA or PSP, suggesting that abnormalities in plasma GlcCer (a MonCer) metabolism are also commonly observed in synucleinopathies, amyloidopathies and tauopathies. (p21, lines 359-371)

2)

Quantitative data are necessary to understand the yield of extraction and the correct determination and comparison within samples.

We agree with your opinion. Thank you for your important remarks.

In present lipidomic analysis, the coefficient of variation (CV) ranged from 4.4 to 9.7%, with a mean of 6.7%. Because of the variability of metabolites values, present method should be compared with healthy subjects for each cohort. Therefore, the relative areas were used as metabolites values in this study.

Based on what you have pointed out, we added sentences in Discussion section. 

Seventh, Quantitative data were not used as metabolites values in this study. In present lipidomic analysis, the coefficient of variation (CV) ranged from 4.4 to 9.7%, with a mean of 6.7%. Because of the variability of metabolites values, present method should be compared with healthy subjects for each cohort [4,5]. Therefore, the relative areas were used as metabolites values in this study. (p24, lines 427-431)

Reviewer #2: General Comment to authors

The authors examine plasma sphingolipid in several neurodegenerative diseases and show striking changes in sphingosine-1-phosphate and GluCer and LacCer species. However, the small sample sizes and the significant age differences in cohort B are concerns, given that age is a significant risk factor in Alzheimer’s disease.

Specific Comments to Authors

1) 

Interestingly, sphingolipids change in all neurodegenerative diseases examined. How do the changes in S1P, GluCer, and LacCer relate to known changes in fluid biomarkers such as plasma amyloid beta 42 and amyloid beta 40 ratio, and plasma ptau levels?. It would be necessary also to measure these fluid biomarkers and not merely rely on MMSE, given the small sample sizes.

Thank you for your suggestion and your important remarks. Since our laboratory could analyze plasma p-tau levels, we analyzed plasma p-tau levels. Pearson Correlation Coefficient was used to correlate plasma p-tau levels and plasma S1P d16.1 levels, plasma S1P d18.1 levels, total plasma MonCer d18:1 levels or total plasma LacCer d18:1 levels in all enrolled groups. Correlation between plasma p-tau levels and plasma S1P d16.1 levels (p = 0.509), plasma S1P d18.1 levels (p = 0.468), plasma MonCer d18:1 levels (p = 0.767), or plasma LacCer d18:1 levels (p = 0.999) showed no correlation.

Based on your suggestion, we added sentences in Results section, Materials and Methods section.

Correlation between plasma p-tau levels and plasma S1P levels, total plasma MonCer levels or total plasma LacCer levels.

To investigate the association between AD-associated protein and sphingolipids, Pearson Correlation Coefficient was used to correlate plasma p-tau levels and plasma S1P d16.1 levels, plasma S1P d18.1 levels, total plasma MonCer d18:1 levels or total plasma LacCer d18:1 levels in all enrolled groups. Correlation between plasma p-tau levels and plasma S1P d16.1 levels (p = 0.509), plasma S1P d18.1 levels (p = 0.468), plasma MonCer d18:1 levels (p = 0.767), or plasma LacCer d18:1 levels (p = 0.999) showed no correlation. (p18, lines 300-307)

Simoa™ Assay

Plasma samples stored at -80°C were thawed and centrifuged at 10,000 x g for 5 minutes. Samples were diluted in advance with the Sample Diluent provided with Assay Kit and applied to the plate. The assay was performed one sample at a time. Simoa™ p-Tau181 Advantage Kit (Quanterix, #103377, Billerica, MA, USA) were used to measure plasma p-Tau181. Measurements were performed according to the instructions for kit. (p12, lines 185-190)

*It was proposed by reviewer 1 to show information on the separation of GalCer from GlcCer. Glucosylceramide (GlcCer) and galactosylceramides (GalCer) are isomers, and MonCer is the sum of both compounds. Although it is difficult to completely separate plasma GalCer and plasma GlcCer from plasma MonCer in present method, it has been shown that the majority of plasma MonCer is composed of plasma GlcCer. In addition, Mielke et al. also described plasma GlcCer as GlcCer (a MonCer) in their PROS ONE manuscript [2]. Thus, we also described plasma GlcCer as GlcCer (a MonCer) in reference to their PROS ONE manuscript.

2) 

How do the plasma changes in sphingolipids relate to brain and CSF changes?

Thank you for your suggestion. We added sentences in Discussion section as you have indicated. 

In previous reports, the increase and decrease in CSF sphingolipids and blood sphingolipids have coincided, suggesting that blood sphingolipids may reflect the dynamics of the central nervous system in neurodegenerative diseases. (p19, lines 322-324)

3) 

The age of the CN population (cohort B) is significantly lower than that of the DLB and AD populations. Since age is a significant risk factor for AD, it is unclear whether the changes are AD-related or age-related. Additional analyses with age as a variable may help tease out whether these data are age-related or pathological.

Thank you for your important remarks. Based on your suggestion, we have performed additional statistical analysis.

Pearson Correlation Coefficient was used to correlate age and plasma S1P d16:1 levels, plasma S1P d18:1 levels, plasma MonCer d18:1 levels, or plasma LacCer d18:1 levels in DLB or AD. In DLB, correlation between age and plasma S1P d16:1 levels (p = 0.544), plasma S1P d18:1 levels (p = 0.644), plasma MonCer d18:1 levels (p = 0.074), or plasma LacCer d18:1 levels (p = 0.315) showed no correlation. In AD, correlation between age and plasma S1P d16:1 levels (p = 0.506), plasma S1P d18:1 levels (p = 0.762), plasma MonCer d18:1 levels (p = 0.434), or plasma LacCer d18:1 levels (p = 0.742) showed no correlation. Therefore, it was inferred that changes in plasma S1P d16:1 levels, plasma S1P d18:1 levels, plasma MonCer d18:1 levels or plasma LacCer d18:1 levels were inferred to be disease-induced changes in AD or DLB.

We added sentences in Discussion section as you have indicated. 

Fifth, cohort B was not an age-matched study. In DLB and AD, correlation analysis between age and plasma S1P d16:1 levels, plasma S1P d18:1 levels, plasma MonCer d18:1 levels, or plasma LacCer d18:1 levels showed no correlation (data not shown). Thus, changes in plasma S1P d16:1 levels, plasma S1P d18:1 levels, plasma MonCer d18:1 levels or plasma LacCer d18:1 levels were inferred to be disease-induced changes in AD or DLB. (p24, lines 421-425)

4) 

What is the recovery of sphingolipids extracted with the SPE column (page 11)? What internal standards were used, and how quantitative were these standards?

Thank you for your suggestion. Based on your suggestion, we added sentences in Materials and Methods section.

The average recovery of sphingolipids extracted with the SPE column is 88% (range 68% to 99.9%). (p11, lines 165-166)

Internal standards are used at a concentration of 50 uM. Cer/Sph Mixture II (Avanti Polar Lipids, LM6005, Birmingham, AL, USA) was used as internal standards of sphingolipid. (p12, lines 181-183)

5) 

Is there a correlation/link between the decrease in S1P and the increase in GlcCer and LacCer in plasma? 

Thank you for your suggestion. We agree with your opinion. Based on your suggestion, we have performed additional statistical analysis.

Pearson Correlation Coefficient was used to correlate plasma MonCer levels and plasma LacCer levels and plasma S1P levels in all enrolled groups. Total plasma MonCer d18:1 levels were significantly positively correlated with total plasma LacCer d18:1 levels (r = 0.5802, p 0.0001). There was no difference between total plasma LacCer d18:1 levels and plasma S1P d16:1 levels (r = -0.0092, p = 0.9091) or plasma S1P d18:1 levels (r = 0.0989, p = 0.2195). There was no difference between total plasma MonCer d18:1 levels and plasma S1P d16:1 levels (r = 0.0719, p = 0.3721), but total plasma MonCer d18:1 levels were significantly slight positively correlated with plasma S1P d18:1 levels (r = 0.2480, p = 0.0028). These results suggest that an increase in plasma MonCer d18:1 levels may be directly related to an increase in LacCer d18:1 levels in all enrolled groups.

We found interesting results, as you pointed out. Thank you very much.

We added sentences in Discussion section, based on the above results. 

Correlation between total plasma MonCer levels and total plasma LacCer levels.

Pearson Correlation Coefficient was used to correlate total plasma MonCer d18:1 levels and total plasma LacCer d18:1 levels in all enrolled groups. Total plasma MonCer d18:1 levels were significantly positively correlated with total plasma LacCer d18:1 levels (r = 0.5802, p 0.0001) (Fig 4) in all enrolled groups. These results suggest that an increase in plasma MonCer may be directly related to an increase in LacCer in all enrolled groups.

Fig 4. Correlation between total plasma MonCer levels and total plasma LacCer levels. (A) Total plasma MonCer d18:1 levels were significantly positively correlated with total plasma LacCer d18:1 levels (r = 0.5802, p 0.0001) in all enrolled groups. (p17-18, lines 289-298)

What enzyme pathways may cause the decrease in S1P and the increase in GluCer and LacCer?

Thank you for your suggestion. Based on your suggestion, we added sentences in Discussion section.

GCS is GlcCer synthase, BGTase6 is LacCer synthase, and SPHK is S1P synthase. These indicate that increased GlcCer and LacCer are caused by increased function of GCS and BGTase6, respectively, and decreased S1P is caused by a relative loss of function of SPHK. (p19, lines 319-321)

6) 

The limitations (page 22) should include that this is a small discovery/cross sectional study that can not account for multiple comparisons for several analytes detected in plasma.

Thank you for your suggestion. We added sentences in Discussion section as you have indicated. 

Sixth, this study is a small discovery/cross sectional study that could not account for multiple comparisons for several analytes detected in plasma. (p24, lines 425-427)

1. Xu H, Boucher FR, Nguyen TT, Taylor GP, Tomlinson JJ, Ortega RA, et al. DMS as an orthogonal separation to LC/ESI/MS/MS for quantifying isomeric cerebrosides in plasma and cerebrospinal fluid. J Lipid Res. 2019;60: 200-211.

2. Mielke MM, Maetzler W, Haughey NJ, Bandaru VV, Savica R, Deuschle C, et al. Plasma ceramide and glucosylceramide metabolism is altered in sporadic Parkinson's disease and associated with cognitive impairment: a pilot study. PLoS One. 2013;8: e73094.

3. Savica R, Murray ME, Persson XM, Kantarci K, Parisi JE, Dickson DW, et al. Plasma sphingolipid changes with autopsy-confirmed Lewy Body or Alzheimer's pathology. Alzheimers Dement (Amst). 2016;3: 43-50.

4. Mori A, Ishikawa KI, Saiki S, Hatano T, Oji Y, Okuzumi A, et al. Plasma metabolite biomarkers for multiple system atrophy and progressive supranuclear palsy. PLoS One. 2019;14: e0223113.

5. Saiki S, Sasazawa Y, Fujimaki M, Kamagata K, Kaga N, Taka H, et al. A metabolic profile of polyamines in parkinson disease: A promising biomarker. Ann Neurol. 2019;86: 251-263.

---

## [Decision Letter · Decision Letter 1]

17 Oct 2022

PONE-D-22-15953R1Plasma sphingolipid abnormalities in neurodegenerative diseasesPLOS ONE

Dear Dr. Takeda,

Thank you for submitting your manuscript to PLOS ONE. After careful consideration, we feel that it has merit but does not fully meet PLOS ONE’s publication criteria as it currently stands. Therefore, we invite you to submit a revised version of the manuscript that addresses the points raised during the review process.

1) Is the manuscript technically sound, and do the data support the conclusions?In response to this question, reviewer #3 says "partly" while reviewer #1 had no response. 2) Have the authors made all data underlying the findings in their manuscript fully available?Reviewer #3 says "no." 3) Reviewer #1 says that his comments pertaining to the original manuscript were not addressed:

(i) I believe that more important would be the knowledge on the others, not considered sphingolipids.

Editor: Point 3(i) has not been addressed, and this was also brought up by reviewer #3.

Information on the separation of GalCer from GlcCer is necessary to understand to which compound are referred the results. Editor: I accept your explanation on the difficulties in separating GalCer from GlcCer.

(iii) Quantitative data are necessary to understand the yield of extraction and the correct determination and comparison within samples.  

Editor: The reviewer was not satisfied as to your response to 3(iii).

 4)  Reviewer#3 comments:(i)The authors do not provide an explanation of which enzyme pathways can be affected in the observed alterations. It would provide important insights if the authors can perform additional experiments in testing the protein levels of the enzymes involved in the pathways. Editor: Please respond to 4(i). Testing protein levels of the enzymes could be helpful. Can this be done?

(ii) What is the outcome of other lipids that were analyzed in this lipidomics analyses? For completeness, they should also include those, even if no differences were observed. If no other lipids were analyzed, the authors should include a reasoning why they only tested these lipids. Editor: You should include the results for other lipids tested.

(iii) The collection of probands remains on very low, which lowers the impact of the manuscript.

Editor:  Please respond to this comment.

(iv) The authors should be careful in claiming that the SL are similar in plasma compared to CSF or brain as was questioned by a reviewer; the papers the authors refer to are indirect and one even identified different findings in serum versus CSF. Perhaps it is better to include this in the limitations that these findings need to be confirmed in CSF or brain for further validation.Editor: please address this comment.

(v) In addition, a few other questions are still open. Did the authors check for different lipid species as different chain length is important to define their function and hence a more in-depth analyses provides a better understanding of the findings.Editor: please address this question.

(vi) Can the authors test if the IPD samples contain mutations in LRRK2 and SNCA as these are regularly identified in IPD and may provide a link to genetic forms of PDEditor: Can you test for these mutations; doing so would definitely increase the impact of your paper. 5) Editor: lines 427-431, revised: "Seventh, Quantitative data were not used as metabolites values in

this study. In present lipidomic analysis, the coefficient of variation (CV) ranged from 4.4 to

9.7%, with a mean of 6.7%. Because of the variability of metabolites values, present method

should be compared with healthy subjects for each cohort. Therefore, the relative areas

were used as metabolites values in this study."

Editor: The above explanation is very hard to understand.

If applicable, we recommend that you deposit your laboratory protocols in protocols.io to enhance the reproducibility of your results. Protocols.io assigns your protocol its own identifier (DOI) so that it can be cited independently in the future. For instructions see: https://journals.plos.org/plosone/s/submission-guidelines#loc-laboratory-protocols. Additionally, PLOS ONE offers an option for publishing peer-reviewed Lab Protocol articles, which describe protocols hosted on protocols.io. Read more information on sharing protocols at https://plos.org/protocols?utm_medium=editorial-emailutm_source=authorlettersutm_campaign=protocols.

We look forward to receiving your revised manuscript.

Kind regards,

Stephan N. Witt, Ph.D.

Academic Editor

PLOS ONE

Reviewers' comments:

Reviewer's Responses to Questions

**Comments to the Author**

1. If the authors have adequately addressed your comments raised in a previous round of review and you feel that this manuscript is now acceptable for publication, you may indicate that here to bypass the “Comments to the Author” section, enter your conflict of interest statement in the “Confidential to Editor” section, and submit your "Accept" recommendation.

Reviewer #1: (No Response)

Reviewer #3: (No Response)

2. Is the manuscript technically sound, and do the data support the conclusions?

Reviewer #1: (No Response)

Reviewer #3: Partly

3. Has the statistical analysis been performed appropriately and rigorously? 

Reviewer #1: (No Response)

Reviewer #3: Yes

4. Have the authors made all data underlying the findings in their manuscript fully available?

Reviewer #1: (No Response)

Reviewer #3: No

5. Is the manuscript presented in an intelligible fashion and written in standard English?

Reviewer #1: (No Response)

Reviewer #3: Yes

6. Review Comments to the Author

Reviewer #1: I requested additional results, but the Authors did not add any technical new result. According to this I maintain my previous comments.

Reviewer #3: Oizumi et al., performed a lipidomics analyses on plasma of patients suffering from different neurodegenerative disorders, including idiopathic Parkinson’s disease and dementia with Lewy bodies. The lipidomics focused on sphingolipidomics and revealed S1P, MonCer, and LacCer to be altered in the plasma from patients suffering from neurodegenerative diseases compared to control groups.

In this revised manuscript the authors do not fully address the questions raised by the reviewers for instance:

The authors do not provide an explanation of which enzyme pathways can be affected in the observed alterations. It would provide important insights if the authors can perform additional experiments in testing the protein levels of the enzymes involved in the pathways

What is the outcome of other lipids that were analyzed in this lipidomics analyses? For completeness, they should also include those, even if no differences were observed. If no other lipids were analyzed, the authors should include a reasoning why they only tested these lipids.

The collection of probands remains on very low, which lowers the impact of the manuscript.

The authors should be careful in claiming that the SL are similar in plasma compared to CSF or brain as was questioned by a reviewer; the papers the authors refer to are indirect and one even identified different findings in serum versus CSF. Perhaps it is better to include this in the limitations that these findings need to be confirmed in CSF or brain for further validation.

In addition, a few other questions are still open. Did the authors check for different lipid species as different chain length is important to define their function and hence a more in-depth analyses provides a better understanding of the findings.

Can the authors test if the IPD samples contain mutations in LRRK2 and SNCA as these are regularly identified in IPD and may provide a link to genetic forms of PD

7. PLOS authors have the option to publish the peer review history of their article (what does this mean?). If published, this will include your full peer review and any attached files.

Reviewer #1: No

Reviewer #3: No

---

## [Author Response · Author response to Decision Letter 1]

14 Nov 2022

Thank you for submitting your manuscript to PLOS ONE. After careful consideration, we feel that it has merit but does not fully meet PLOS ONE’s publication criteria as it currently stands. Therefore, we invite you to submit a revised version of the manuscript that addresses the points raised during the review process.

1) Is the manuscript technically sound, and do the data support the conclusions?

In response to this question, reviewer #3 says "partly" while reviewer #1 had no response.

Thanks for the suggestion. We have made the new revised manuscript to improve on original manuscript.

2) Have the authors made all data underlying the findings in their manuscript fully available?

Reviewer #3 says "no."

Thank you for your suggestion. Supporting information (S1 Table, S2 Table, and S3 Table) and additional Table (Table 2 and Table 3) has been added to the new revised manuscript to improve on original manuscript.

3) Reviewer #1 says that his comments pertaining to the original manuscript were not addressed:

(i) I believe that more important would be the knowledge on the others, not considered sphingolipids.

Editor: Point 3(i) has not been addressed, and this was also brought up by reviewer #3.

Thank you for your suggestion. We have described a comprehensive knowledge of lipid abnormalities reported in neurodegenerative diseases.

We added sentences in Discussion section. Changes in the revised manuscript are highlighted in yellow.

Recessive mutations in the GBA1 (glucocerebrosidase) gene cause Gaucher disease. Heterozygous GBA1 mutation carriers exhibit much greater incidence of PD than the general population [1,2]. Likewise, mutations in the NPC1 (NPC intracellular cholesterol transporter 1) and SMPD1 (sphingomyelin phosphodiesterase 1) genes, which cause Niemann-Pick disease, have been shown to be risk genes for IPD [3,4]. One of the phospholipase A2 members, PLA2G6 or iPLA2-VIA/iPLA2β, has been isolated as the gene responsible for an autosomal recessive form of PD linked to the PARK14 locus [5]. Compared to the most common e3 isoform, the e4 isoform of ApoE (ApoE4) is the strongest genetic risk factor for late-onset AD [6]. β amyloid accumulation in NPC1 (NPC intracellular cholesterol transporter 1) gene, which cause Niemann-Pick type C, mutant cells and NPC mouse brain suggests the association between cholesterol metabolism and AD [7]. As described, several lipid-related genes have been reported as risk genes or causative genes in AD and IPD. In addition, various lipid abnormalities have been reported in IPD and AD, such as fatty acids, glycerolipids, glycerophospholipids, sphingolipids, sterols, and lipoproteins [8,9]. However, it is still unclear which lipid metabolism abnormalities play the most important role in neurodegenerative diseases. Plasma lipidomics is an unbiased method and can find important lipids in neurodegenerative diseases. For this reason, plasma lipidomics was performed in neurodegenerative diseases in this study. (p25, lines 363-379)

(ii) Information on the separation of GalCer from GlcCer is necessary to understand to which compound are referred the results. 

Editor: I accept your explanation on the difficulties in separating GalCer from GlcCer.

Thank you for your acceptance.

(iii) Quantitative data are necessary to understand the yield of extraction and the correct determination and comparison within samples. 

Editor: The reviewer was not satisfied as to your response to 3(iii).

Thank you for your suggestion. We measured relative areas as quantitative value based on PROS ONE manuscript [10]. Lipidomics has the variability of metabolite values in each study. For this reason, each metabolite should be normalized based on the internal standard (IS) level and sample volume. The normalized each metabolite was represented as relative area and used as the quantitative value. Additionally, information on IS was described in Materials and Methods section.

We changed sentence in Materials and Methods, and Discussion section. Changes in the revised manuscript are highlighted in yellow.

Target metabolites are divided into categories (fatty acids, acylcarnitines, oxylipins, lysophospholipids, platelet-activating factors, glycosphingolipids, sphinganines, sphingosines, and steroids) according to their physical properties, and the recovery rate is corrected using the corresponding IS (Internal standards). Based on these reports, these IS were selected [11-13]. The recovery rate of analytes during extraction ranged from 68% to 129%, with a mean of 96%. IS coefficient of variation ranged from 4.4 to 9.7%, with a mean of 6.7%. The peak area of each metabolite was then normalized based on IS level and sample volume for relative quantification. The normalized each metabolite was represented as relative area and used as the quantitative value based on previous reports [10,14]. (p12, lines 182-191)

Seventh, relative area was used in this study as the quantitative value for each metabolite based on previous reports [10,14]. Lipidomics has the variability of metabolite values in each study. For this reason, each metabolite should be normalized based on the IS level and sample volume. The normalized each metabolite was represented as relative area and used as the quantitative value. (p32-33, lines 518-522)

4) Reviewer#3 comments:

(i)The authors do not provide an explanation of which enzyme pathways can be affected in the observed alterations. It would provide important insights if the authors can perform additional experiments in testing the protein levels of the enzymes involved in the pathways.

Editor: Please respond to 4(i). Testing protein levels of the enzymes could be helpful. Can this be done?

Thank you for your suggestion. However, in this study we cannot perform additional experiments in testing the protein levels of the enzymes involved in sphingolipid pathways.

We added sentences in Discussion section. Changes in the revised manuscript are highlighted in yellow.

Sixth, in this study the protein levels of the enzymes involved in sphingolipid pathways were not evaluated in all enrolled patients. The protein levels of the enzymes involved in sphingolipid pathways need to be performed in future studies. (p32, lines 516-518)

(ii) What is the outcome of other lipids that were analyzed in this lipidomics analyses? For completeness, they should also include those, even if no differences were observed. If no other lipids were analyzed, the authors should include a reasoning why they only tested these lipids.

Editor: You should include the results for other lipids tested.

Thank you for your suggestion. We included the results for other lipids tested.

We added sentences in Results, Discussion, Supporting information section. We added S1 Table and S2 Table. Changes in the revised manuscript are highlighted in yellow.

Plasma other lipid metabolite levels in neurodegenerative diseases.

Plasma other lipid metabolite (other sphingolipids, sphinganines, gangliosides, free fatty acids, acylcarnitnes, lysophospholipids, platelet-activating factor, acylethanolamine, thyroid hormone, cholic acids, and steroids) levels were compared between the CN group and the IPD, DLB, MSA, AD and PSP groups. Oxylipins were not statistically analyzed because it is considered unsuitable for statistical analysis due to the large number of undetectable samples. Statistical significance was examined using one-tailed Welch’s t tests. Plasma ceramide-1-phosphate (C1P) levels were significantly higher in the PD, DLB, and AD groups versus the control group (S1 table). Plasma GM3 ganglioside and GD3 ganglioside levels were significantly higher in all neurodegenerative disease groups (IPD, DLB, MSA, AD, and PSP) versus the control group (S1 table). Plasma lysophosphatidic acid, lysophosphatidylcholine, lysophosphatidylethanolamine, lysophosphatidylglycerol, lysophosphatidylserine levels were lower in DLB group versus the control group (S1 table). Plasma cortisone levels were significantly higher in the PD, MSA and PSP groups versus the control group (S2 Table). (p19-20, lines 332-345)

In this study, we found that plasma GM3 and GD3 ganglioside levels were significantly higher in the neurodegenerative disease groups than in the CN group. Gangliosides are lipids classified as sphingolipids. GM3 ganglioside is the starting material for gangliosides, which are biosynthesized by the binding of sialic acid to LacCer [15,16]. Previously, plasma GM3 ganglioside levels have been shown to be elevated in PD [17]. The elevated plasma GM3 ganglioside levels in individuals with IPD in this study were in good accordance with the results of previous study. GD3 ganglioside is the gangliosides, which are biosynthesized by the binding of sialic acid to GM3 ganglioside [15,16]. GM3 and GD3 gangliosides are components of lipid rafts and are implicated in cell death [18,19]. Abnormalities in lipid rafts are also considered to be one of the major causes of neurodegenerative diseases [20]. Homozygous knockout mice for B4galnt1, a ganglioside synthase, have been shown to exhibit PD-like motor deficits and cause dopaminergic neuron degeneration [21]. Taken together, these results suggest that elevated plasma GM3 and GD3 gangliosides may reflect abnormal lipid rafts in neurodegenerative diseases. In this study, we found that plasma C1P levels were significantly higher in the IPD, DLB, and AD groups than in the CN group. C1P is classified as a sphingolipid, a lipid mainly involved in cell survival and inflammation [22,23]. Neuroinflammation is also considered to be a one of the major causes in PD, DLB and AD [24-26]. Therefore, elevated C1P may reflect neuroinflammation in these diseases. (p30-31, lines 472-489)

S1 Table. Plasma other sphinogolipids, sphinganines, gangliosides, free fatty acids, acylcarnitnes, lysophospholipids levels in neurodegenerative diseases.

Statistical methods: The metabolite level ratio of IPD, DLB, MSA, AD, or PSP to CNs. Statistical significance was examined using one-tailed Welch's t tests (P 0.05).

Abbreviations: ceramide-1-phosphate (C1P), sphinganine-1-phosphate (SG1P), lysophosphatidic acid (LPA), lysophosphatidylcholine (LPC), lysophosphatidylethanolamine (LPE), lysophosphatidylglycerol (LPG), lysophosphatidylinositol (LPI), lysophosphatidylserine (LPS) (p46, lines 758-764)

S2 Table. Plasma platelet-activating factor, acylethanolamine, thyroid hormone, cholic acids, steroids levels in neurodegenerative diseases.

Statistical methods: The metabolite level ratio of IPD, DLB, MSA, AD, or PSP to CNs. Statistical significance was examined using one-tailed Welch's t tests (P 0.05).

Abbreviations: platelet-activating factor (PAF) (p46, lines 765-769)

 

(iii) The collection of probands remains on very low, which lowers the impact of the manuscript.

Editor: Please respond to this comment.

Thank you for your suggestion. We considered the collection of probands to be genetic analysis.

We could not test GBA, LRRK2, and SNCA mutations in this study, so we changed sentences in Discussion section. Changes in the revised manuscript are highlighted in yellow.

First, Analysis the major causative genes or risk genes of PD during lipidomics were not evaluated. GBA1 mutations were not evaluated in all enrolled IPD patients. Based on the GBA1 genotype and clinical analysis, it has been reported that GBA1 mutation is the most common genetic risk factor for IPD patients, accounting for as many as 7% of all IPD patients in multicenter analyses [1,2]. On the other hand, only approximately 3% of Asian IPD patients with no apparent family history of parkinsonism are GBA1 mutation carriers [27]. IPD in GBA1 mutation carriers generally has an early onset [28]. However, there was no apparent family history of parkinsonism or dementia in all enrolled IPD patients, with a later mean age of onset in the enrolled IPD patients that was 67.2 years in cohort A and 65.2 years in cohort C. Elevated plasma GlcCer levels have recently been reported in GBA1 mutation carriers of IPD. Elevated plasma GlcCer levels have also been reported in non-GBA1 mutation carriers of IPD. These indicate that elevated plasma GlcCer is found in IPD with or without GBA mutation. Taken together, it is not plausible that a GBA1 mutation did not significantly affect elevated plasma GlcCer (a MonCer) levels in the IPD patients in this study. In addition, in this study LRRK2 and SNCA mutations, the major causative genes of PD, were not evaluated in all enrolled IPD patients. Analysis the major causative genes or risk genes of PD during lipidomics need to be performed in future studies. (p31-32, lines 491-507)

In addition, we considered the collection of probands to be the number of cases, so we changed sentences in Discussion section. 

Second, this study is a small cases and cross-sectional study that could not account for multiple comparisons for several analytes detected in plasma. Future additional cases and longitudinal studies need to be performed. (p32, lines 507-509)

(iv) The authors should be careful in claiming that the SL are similar in plasma compared to CSF or brain as was questioned by a reviewer; the papers the authors refer to are indirect and one even identified different findings in serum versus CSF. Perhaps it is better to include this in the limitations that these findings need to be confirmed in CSF or brain for further validation.

Editor: please address this comment.

Thank you for your suggestion. We changed sentence in Discussion section as you indicated. Changes in the revised manuscript are highlighted in yellow.

Eighth, the increase and decrease in CSF sphingolipids and blood sphingolipids have coincided [30,31,58,59] in previous reports. On the other hand, one report even identified different findings in serum versus CSF [29,30]. These reports are indirect and sphingolipids need to be confirmed in CSF or brain for further validation. (p33, lines 523-526)

(v) In addition, a few other questions are still open. Did the authors check for different lipid species as different chain length is important to define their function and hence a more in-depth analyses provides a better understanding of the findings.

Editor: please address this question.

Thank you for your suggestion. To check for different lipid species as different chain length, we examined the association between lipid abnormalities and chain length in MonCers and LacCers.

We added sentences in Results section and additional Table (Table 2 and Table 3). Changes in the revised manuscript are highlighted in yellow.

We compared MonCer (d18:1/14:0), MonCer (d18:1/16:0), MonCer (d18:1/16:1), MonCer (d18:1/18:0), MonCer (d18:1/18:1), MonCer (d18:1/20:0), MonCer (d18:1/20:1), MonCer (d18:1/22:0), MonCer (d18:1/22:1), MonCer (d18:1/22:2), MonCer (d18:1/24:0), MonCer (d18:1/24:1), and MonCer (d18:1/24:2) between the CN group and the IPD, DLB, MSA, AD, or PSP groups (Table 2). The χ-square test was used to examine the association between lipid abnormalities and chain length in MonCer d18:1. No statistically significant difference was found between lipid abnormalities and chain length (P = 0.5522) in all enrolled groups. (p16, lines 261-268)

We compared LacCer (d18:1/14:0), LacCer (d18:1/16:0), LacCer (d18:1/16:1), LacCer (d18:1/18:0), LacCer (d18:1/18:1), LacCer (d18:1/20:0), LacCer (d18:1/20:1), LacCer (d18:1/22:0), LacCer (d18:1/22:1), LacCer (d18:1/22:2), LacCer (d18:1/24:0), LacCer (d18:1/24:1), and LacCer (d18:1/24:2) between the CN group and the IPD, DLB, MSA, AD, or PSP groups (Table 3). The χ-square test was used to examine the association between lipid abnormalities and chain length in LacCers d18:1. No statistically significant difference was found between lipid abnormalities and chain length (P = 0.5522) in all enrolled groups. (p17-18, lines 296-302)

  

(vi) Can the authors test if the IPD samples contain mutations in LRRK2 and SNCA as these are regularly identified in IPD and may provide a link to genetic forms of PD

Editor: Can you test for these mutations; doing so would definitely increase the impact of your paper.

Thank you for your suggestion. However, we could not test LRRK2, and SNCA mutations in this study.

We added sentences in Discussion section. Changes in the revised manuscript are highlighted in yellow.

In addition, in this study LRRK2 and SNCA mutations, the major causative genes of PD, were not evaluated in all enrolled IPD patients. Analysis the major causative genes or risk genes of PD during lipidomics need to be performed in future studies. (p32, lines 504-507)

5) Editor: lines 427-431, revised: "Seventh, Quantitative data were not used as metabolites values in this study. In present lipidomic analysis, the coefficient of variation (CV) ranged from 4.4 to 9.7%, with a mean of 6.7%. Because of the variability of metabolites values, present method should be compared with healthy subjects for each cohort. Therefore, the relative areas were used as metabolites values in this study."

Editor: The above explanation is very hard to understand.

Thank you for your suggestion. We agree with your opinion. We measured relative areas as quantitative value based on PROS ONE manuscript [10]. Lipidomics has the variability of metabolite values in each study. For this reason, each metabolite should be normalized based on the internal standard (IS) level and sample volume. The normalized each metabolite was represented as relative area and used as the quantitative value.

We changed sentence in Discussion section as you indicated. Changes in the revised manuscript are highlighted in yellow.

Seventh, relative area was used in this study as the quantitative value for each metabolite based on previous reports [10,14]. Lipidomics has the variability of metabolite values in each study. For this reason, each metabolite should be normalized based on the IS level and sample volume. The normalized each metabolite was represented as relative area and used as the quantitative value. (p32-33, lines 518-522)

1. Sidransky E, Nalls MA, Aasly JO, Aharon-Peretz J, Annesi G, Barbosa ER, et al. Multicenter analysis of glucocerebrosidase mutations in Parkinson's disease. N Engl J Med. 2009;361: 1651-1661.

2. Lesage S, Anheim M, Condroyer C, Pollak P, Durif F, Dupuits C, et al. Large-scale screening of the Gaucher's disease-related glucocerebrosidase gene in Europeans with Parkinson's disease. Hum Mol Genet. 2011;20: 202-210.

3. Foo JN, Liany H, Bei JX, Yu XQ, Liu J, Au WL, et al. Rare lysosomal enzyme gene SMPD1 variant (p.R591C) associates with Parkinson's disease. Neurobiol Aging. 2013;34: 2890 e2813-2895.

4. Kluenemann HH, Nutt JG, Davis MY, Bird TD. Parkinsonism syndrome in heterozygotes for Niemann-Pick C1. J Neurol Sci. 2013;335: 219-220.

5. Gregory A, Westaway SK, Holm IE, Kotzbauer PT, Hogarth P, Sonek S, et al. Neurodegeneration associated with genetic defects in phospholipase A(2). Neurology. 2008;71: 1402-1409.

6. Chartier-Harlin MC, Parfitt M, Legrain S, Pérez-Tur J, Brousseau T, Evans A, et al. Apolipoprotein E, epsilon 4 allele as a major risk factor for sporadic early and late-onset forms of Alzheimer's disease: analysis of the 19q13.2 chromosomal region. Hum Mol Genet. 1994;3: 569-574.

7. Yamazaki T, Chang TY, Haass C, Ihara Y. Accumulation and aggregation of amyloid beta-protein in late endosomes of Niemann-pick type C cells. J Biol Chem. 2001;276: 4454-4460.

8. Yin F. Lipid metabolism and Alzheimer's disease: clinical evidence, mechanistic link and therapeutic promise. FEBS J. 2022.

9. Xicoy H, Wieringa B, Martens GJM. The Role of Lipids in Parkinson's Disease. Cells. 2019;8.

10. Mori A, Ishikawa KI, Saiki S, Hatano T, Oji Y, Okuzumi A, et al. Plasma metabolite biomarkers for multiple system atrophy and progressive supranuclear palsy. PLoS One. 2019;14: e0223113.

11. Hayasaka R, Tabata S, Hasebe M, Ikeda S, Ohnuma S, Mori M, et al. Metabolomic Analysis of Small Extracellular Vesicles Derived from Pancreatic Cancer Cells Cultured under Normoxia and Hypoxia. Metabolites. 2021;11.

12. Suzuki Y, Hayasaka R, Hasebe M, Ikeda S, Soga T, Tomita M, et al. Comparative Metabolomics of Small Molecules Specifically Expressed in the Dorsal or Ventral Marginal Zones in Vertebrate Gastrula. Metabolites. 2022;12.

13. Ikeda K. Mass Spectrometric Analysis of Phospholipids by Target Discovery Approach. 2015. pp. 349-356.

14. Saiki S, Sasazawa Y, Fujimaki M, Kamagata K, Kaga N, Taka H, et al. A metabolic profile of polyamines in parkinson disease: A promising biomarker. Ann Neurol. 2019;86: 251-263.

15. Merrill AH, Jr. Sphingolipid and glycosphingolipid metabolic pathways in the era of sphingolipidomics. Chem Rev. 2011;111: 6387-6422.

16. Yamaji T, Hanada K. Sphingolipid metabolism and interorganellar transport: localization of sphingolipid enzymes and lipid transfer proteins. Traffic. 2015;16: 101-122.

17. Chan RB, Perotte AJ, Zhou B, Liong C, Shorr EJ, Marder KS, et al. Elevated GM3 plasma concentration in idiopathic Parkinson's disease: A lipidomic analysis. PLoS One. 2017;12: e0172348.

18. Sohn H, Kim YS, Kim HT, Kim CH, Cho EW, Kang HY, et al. Ganglioside GM3 is involved in neuronal cell death. FASEB J. 2006;20: 1248-1250.

19. De Maria R, Lenti L, Malisan F, d'Agostino F, Tomassini B, Zeuner A, et al. Requirement for GD3 ganglioside in CD95- and ceramide-induced apoptosis. Science. 1997;277: 1652-1655.

20. Schengrund CL. Lipid rafts: keys to neurodegeneration. Brain Res Bull. 2010;82: 7-17.

21. Wu G, Lu ZH, Kulkarni N, Amin R, Ledeen RW. Mice lacking major brain gangliosides develop parkinsonism. Neurochem Res. 2011;36: 1706-1714.

22. Chalfant CE, Spiegel S. Sphingosine 1-phosphate and ceramide 1-phosphate: expanding roles in cell signaling. J Cell Sci. 2005;118: 4605-4612.

23. Arana L, Gangoiti P, Ouro A, Trueba M, Gomez-Munoz A. Ceramide and ceramide 1-phosphate in health and disease. Lipids Health Dis. 2010;9: 15.

24. Leng F, Edison P. Neuroinflammation and microglial activation in Alzheimer disease: where do we go from here? Nat Rev Neurol. 2021;17: 157-172.

25. Gelders G, Baekelandt V, Van der Perren A. Linking Neuroinflammation and Neurodegeneration in Parkinson's Disease. J Immunol Res. 2018;2018: 4784268.

26. Surendranathan A, Rowe JB, O'Brien JT. Neuroinflammation in Lewy body dementia. Parkinsonism Relat Disord. 2015;21: 1398-1406.

27. Wu YR, Chen CM, Chao CY, Ro LS, Lyu RK, Chang KH, et al. Glucocerebrosidase gene mutation is a risk factor for early onset of Parkinson disease among Taiwanese. J Neurol Neurosurg Psychiatry. 2007;78: 977-979.

28. Sato C, Morgan A, Lang AE, Salehi-Rad S, Kawarai T, Meng Y, et al. Analysis of the glucocerebrosidase gene in Parkinson's disease. Mov Disord. 2005;20: 367-370.

29. Mielke MM, Maetzler W, Haughey NJ, Bandaru VV, Savica R, Deuschle C, et al. Plasma ceramide and glucosylceramide metabolism is altered in sporadic Parkinson's disease and associated with cognitive impairment: a pilot study. PLoS One. 2013;8: e73094.

30. Huh YE, Park H, Chiang MSR, Tuncali I, Liu G, Locascio JJ, et al. Glucosylceramide in cerebrospinal fluid of patients with GBA-associated and idiopathic Parkinson's disease enrolled in PPMI. NPJ Parkinsons Dis. 2021;7: 102.

---

## [Decision Letter · Decision Letter 2]

29 Nov 2022

PONE-D-22-15953R2Plasma sphingolipid abnormalities in neurodegenerative diseasesPLOS ONE

Dear Dr. Takeda,

Thank you for submitting your manuscript to PLOS ONE. Please make the minor corrections to table S1, and resubmit with no other changes. See my comments below.

We look forward to receiving your revised manuscript.

Kind regards,

Stephan N. Witt, Ph.D.

Academic Editor

PLOS ONE

Journal Requirements:

Additional Editor Comments :

I will accept your manuscript after you make corrections to S1 Table. In S1 Table, you repeatedly misspell the word "sphingolipids." Throughout the table sphingolipids is spelled "sphinogolipids." Please fix.

Reviewers' comments:

Reviewer's Responses to Questions

**Comments to the Author**

1. If the authors have adequately addressed your comments raised in a previous round of review and you feel that this manuscript is now acceptable for publication, you may indicate that here to bypass the “Comments to the Author” section, enter your conflict of interest statement in the “Confidential to Editor” section, and submit your "Accept" recommendation.

Reviewer #1: (No Response)

Reviewer #3: All comments have been addressed

2. Is the manuscript technically sound, and do the data support the conclusions?

Reviewer #1: Partly

Reviewer #3: Yes

3. Has the statistical analysis been performed appropriately and rigorously? 

Reviewer #1: N/A

Reviewer #3: Yes

4. Have the authors made all data underlying the findings in their manuscript fully available?

Reviewer #1: Yes

Reviewer #3: Yes

5. Is the manuscript presented in an intelligible fashion and written in standard English?

Reviewer #1: Yes

Reviewer #3: Yes

6. Review Comments to the Author

Reviewer #1: The human plasma sphingolipid pattern has been described using different methodological approaches. It would be important to understand which are the changes of sphingolipid pattern in neurodegenerative diseases. At least a simple TLC procedure could give the preliminary information.

GlcCer and GalCer sphingolipids are the result of completely different metabolic pathways. In my opinion it is necessary to know the behavior of these two compounds. There are several procedures that allow their separation.

Reviewer #3: The authors have sufficiently addressed my comments and added profound explanation to unresolved questions that they also included in the manuscript.

7. PLOS authors have the option to publish the peer review history of their article (what does this mean?). If published, this will include your full peer review and any attached files.

Reviewer #1: No

Reviewer #3: No

---

## [Author Response · Author response to Decision Letter 2]

2 Dec 2022

Additional Editor Comments :

I will accept your manuscript after you make corrections to S1 Table. In S1 Table, you repeatedly misspell the word "sphingolipids." Throughout the table sphingolipids is spelled "sphinogolipids." Please fix.

Thank you for your suggestion. We have changed the misspelled "sphinogolipids." in S1 Table and Supporting information to correct "sphingolipids." Changes in the revised manuscript are highlighted in yellow.

S1 Table. Plasma other sphingolipids, sphinganines, gangliosides, free fatty acids, acylcarnitnes, lysophospholipids levels in neurodegenerative diseases. (p47, lines 758-759)

---

## [Editor Report · Decision Letter 3]

5 Dec 2022

Plasma sphingolipid abnormalities in neurodegenerative diseases

PONE-D-22-15953R3

Dear Dr. Takeda,

We’re pleased to inform you that your manuscript has been judged scientifically suitable for publication and will be formally accepted for publication once it meets all outstanding technical requirements.

Kind regards,

Stephan N. Witt, Ph.D.

Academic Editor

PLOS ONE
---

## [Editor Report · Acceptance letter]

8 Dec 2022

PONE-D-22-15953R3 

Plasma sphingolipid abnormalities in neurodegenerative diseases 

Dear Dr. Takeda:

I'm pleased to inform you that your manuscript has been deemed suitable for publication in PLOS ONE. Congratulations! Your manuscript is now with our production department. 

Kind regards, 

on behalf of

Dr. Stephan N. Witt 

Academic Editor

PLOS ONE